# Cache Coherent Resampling for Efficient Test Time Scaling in LLM Reasoning via Adaptive Sequential Monte Carlo

**Ke Wang** [1]  **Zehao Yu** [2]  **Luwei Wang** [1]  **Yongchao Huang** [3]

## Abstract

Recent work shows that chain based sampling for power shaped trajectory distributions can deliver large test time gains from a fixed base LLM and can approach RL trained reasoners such as GRPO. Deployment is the bottleneck. Autoregressive Metropolis Hastings is inherently serial, limits GPU utilization, and exhibits extreme tail latency at high budgets, reaching p95 = 1318s on MATH500 at $128\times$. We propose Adaptive Sequential Monte Carlo (ASMC)[1] , a parallel particle inference method that targets power shaped trajectory distributions while adapting particle populations to problem hardness. To make resampling practical for Transformers, we introduce cache coherent resampling, which realizes ancestry updates by reordering KV caches and other particle bound tensors, avoiding prefix recomputation. On MATH500 at the same budget, ASMC attains 80.6% exact-match accuracy with p95 = 73.7s, substantially reducing the tail latency of sequential MCMC and providing additional high-accuracy operating points beyond the saturation of best-of-n. We further analyze particle degeneracy and find that collapse severity, measured by low $\text{ESS}_{\min}/N$, strongly predicts failures, while sensitivity to the resampling scheme is limited.

## 1. Introduction

Test time scaling allocates additional inference compute to improve reasoning performance without retraining. Common approaches include independent sampling with aggregation, such as best of $n$ and self-consistency, and trajectory level sampling methods that target sharpened distributions over long autoregressive sequences. Recent work argues that sampling from a power-shaped trajectory distribution, where the joint probability of a sequence is raised to a power $\alpha > 1$, can yield substantial gains from a fixed base model and can approach RL-trained reasoners such as GRPO without additional training (Shao et al., 2024; Karan & Du, 2025). Practical implementations typically rely on chain-based Metropolis-Hastings to approximate this target (Metropolis et al., 1953), which is algorithmically appealing but fundamentally sequential.

The bottleneck is tail latency. This creates a systems–algorithm mismatch: chain-based inference[2] runs in a sequential decode loop, cannot exploit batching effectively, and yields tail latency that dominates user-facing cost. On MATH500, under a $128\times$ budget, sequential MCMC reaches p95 = 1318s. Furthermore, compute proxies can be miscalibrated across methods because they shift work between high-throughput prefill or teacher forcing and low-throughput autoregressive decoding. We therefore emphasize wall clock latency, especially p95, for deployment facing comparisons, while using interaction-based accounting for controlled compute matched selection and interpretability.

A second obstacle arises from Transformer inference itself: efficient decoding relies on per-layer KV caches. Resampling in SMC reassigns particle ancestries, requiring *cache-coherent* updates to KV caches and all particle-bound state. Naively rebuilding caches by replaying prefixes requires replaying $O(NL)$ tokens and incurs $O(NL^2)$ attention work, which can exceed memory or latency limits for long contexts. To mitigate this, we introduce **cache-coherent resampling** via state reordering: we apply the ancestor mapping as a gather operation across the particle dimension for KV caches and all relevant tensors, avoiding prefix recomputation.

**Contributions.** (i) We introduce ASMC, a parallel parti-

[1]School of Informatics, University of Edinburgh, Edinburgh, UK [2]Department of Computer Science, University College London, London, UK [3]School of Natural and Computing Science, University of Aberdeen, Aberdeen, UK. Correspondence to: Ke Wang <K.Wang-72@sms.ed.ac.uk>.

*Proceedings of the $43^{rd}$ International Conference on Machine Learning*, Seoul, South Korea. PMLR 306, 2026. Copyright 2026 by the author(s).

[1]Code is available at our GitHub repository.

---

[2]While both MH and SMC can be viewed as MCMC methods, they are distinguished by their dependency structure. In the following sections, we use 'MCMC' to denote sequential chain-based methods, excluding SMC.

cle inference method for test-time scaling of autoregressive reasoning, with fixed-$N$ and adaptive-$N$ variants. (ii) We propose cache-coherent resampling for Transformers, which reorders KV caches under the ancestor mapping and avoids prefix replay; we include a full-rebuild systems ablation. (iii) We provide unified instrumentation for attention interactions ($C_{\text{int}}$) and batch-steps, and report both compute-matched selections and end-to-end latency (p50/p95) for deployment-facing comparison. (iv) On MATH500, we show that ASMC improves the high-budget accuracy–p95 frontier, substantially reduces the tail latency of sequential MCMC, and provides complementary high-accuracy operating points beyond best-of-n saturation. We also analyze particle degeneracy, finding that low $\text{ESS}_{\min}/N$ predicts failures while sensitivity to the resampling scheme is limited.

## 2. Preliminaries

### 2.1. Power Distribution Shaped Trajectories in LLMs

We consider a reasoning task where an LLM, parameterized by $\theta$, generates a sequence of tokens $x = (x_1, \ldots, x_T)$ from a vocabulary $\mathcal{V}$, conditioned on a context (prompt) $c$. Let $p_\theta(x \mid c) = \prod_{t=1}^{T} p_\theta(x_t \mid c, x_{<t})$ denote the standard autoregressive distribution. Standard decoding methods (e.g., greedy, nucleus sampling (Holtzman et al., 2020)) sample directly from $p_\theta$. However, for complex reasoning, we aim to sample from a *sharpened* target distribution that concentrates probability mass on high-likelihood, coherent trajectories. The *power-shaped* target distribution is defined as (Karan & Du, 2025; Neal, 2001; Friel & Pettitt, 2008):

$$\pi(x \mid c) \propto p_\theta(x \mid c)^\alpha, \quad \alpha > 1. \tag{1}$$

By setting $\alpha > 1$, we amplify the relative probability of high-quality solutions compared to the base model; specifically, for any two sequences $x, x'$ where $p(x) > p(x')$, the ratio $p(x)^\alpha / p(x')^\alpha$ grows exponentially with $\alpha$, effectively suppressing low-likelihood reasoning traces (Karan & Du, 2025).

**Contrast with Low-Temperature Sampling.** It is a common misconception that standard low-temperature sampling (re-scaling logits at each step) samples from the power distribution $p(x)^\alpha$. Low-temperature sampling modifies the *conditional* next-token probability:

$$p_{\text{temp}}(x_t \mid c, x_{<t}) \propto p_\theta(x_t \mid c, x_{<t})^\alpha. \tag{2}$$

This formulation corresponds to an *exponent of sums* (Karan & Du, 2025), as it essentially sharpens the marginalized probability of future completions: $p_{\text{temp}} \propto (\sum_{x_{>t}} p_\theta(x))^\alpha$. However, the true conditional distribution required to sample from the global power target $\pi(x) \propto p(x)^\alpha$ is fundamentally different. By marginalizing over all future completions $x_{>t}$, the conditional probability for the power distribution

is given by:

$$p_{\text{pow}}(x_t \mid c, x_{<t}) = \frac{\sum_{x_{>t}} p_\theta(x_{0:T}|c)^\alpha}{\sum_{x_t'} \sum_{x_{>t}'} p_\theta(x_{0:T}'|c)^\alpha} \propto \sum_{x_{>t}} p_\theta(x_{0:T}|c)^\alpha. \tag{3}$$

The key difference is that the power distribution targets a *sum of exponents* (Eq. 3), aggregating sharpened likelihoods over all future paths. Standard sampling instead optimizes an *exponent of sums*, acting greedily without accounting for how exponentiation reshapes future likelihoods. Standard sampling therefore cannot target $\pi(x)$ accurately, and sequential inference methods such as MH or SMC are required.

### 2.2. Sequential Monte Carlo (SMC)

Sequential Monte Carlo (SMC) (Doucet et al., 2000a) approximates a sequence of distributions $\{\pi_t\}$ using a parallel population of $N$ weighted particles $\{x_{1:t}^{(i)}, w_t^{(i)}\}$. Unlike MCMC chains, SMC evolves the full population simultaneously, enabling batched decoding on GPUs. At each step $t$, SMC proceeds in three stages:

(1) **Propagation**: extend each particle by sampling

$$x_t^{(i)} \sim q_t(\cdot \mid x_{1:t-1}^{(i)});$$

(2) **Reweighting**: update

$$w_t^{(i)} \propto w_{t-1}^{(i)} \cdot \pi_t(x_{1:t}^{(i)}) / [\pi_{t-1}(x_{1:t-1}^{(i)}) \cdot q_t(x_t^{(i)} \mid x_{1:t-1}^{(i)})];$$

(3) **Resampling**: when effective sample size $\text{ESS}_t = (\sum w_t^{(i)})^2 / \sum (w_t^{(i)})^2 < \tau N$, resample to eliminate low-weight particles.

Resampling presents a systems challenge for Transformers: surviving particles must inherit the KV caches of their ancestors. Naive recomputation requires $O(NL)$ token replay and $O(NL^2)$ attention work; Section 4 introduces cache-coherent resampling to resolve this bottleneck.

## 3. Adaptive Sequential Monte Carlo (ASMC) for LLM Reasoning

As illustrated in Fig.1, MH is inherently sequential: it generates a candidate, accepts or rejects it based on an acquisition score, then iterates. This "generate-then-verify" loop introduces two inefficiencies: (1) it suffers from serial execution that underutilizes massive GPU parallelism, and (2) rejected samples waste compute.

To address this system-algorithm mismatch, we propose **Adaptive Sequential Monte Carlo (ASMC)**. As shown in Fig.1, ASMC replaces the single Markov chain with a parallel population of $N$ particles. This transforms the inference problem from a temporal search (waiting for a chain to mix) into a spatial search (evolving a population),

allowing us to scale compute along the batch dimension.

## 3.1. Probabilistic Formulation

We formally define a sequence of prefix targets whose terminal distribution is the sharpened power-shaped target. For a partial trajectory $x_{1:t}$, we define the unnormalized prefix target:

$$\pi_t(x_{1:t} \mid c) \propto p_\theta(x_{1:t} \mid c)^{\alpha^\star} = \prod_{s=1}^{t} p_\theta(x_s \mid c, x_{<s})^{\alpha^\star}, \quad \alpha^\star > 1, \tag{4}$$

where $\alpha^\star$ is the fixed target exponent. Separately, we use an annealed proposal-shaping schedule $\beta_t$, with $\beta_t$ increasing from an exploratory initial value to $\alpha^\star$. This schedule allows the proposal to explore diverse prefixes early on before concentrating on high-likelihood reasoning paths as the sequence lengthens, while the target exponent itself remains fixed.

At each step $t$, particles are extended, i.e., the next token is added to the existing sequence, as per a *proposal distribution* $q_t(x_t \mid c, x_{<t})$. To ensure robustness against mode collapse, we employ a defensive mixture proposal:

$$q_t = (1 - \epsilon) \, q_t^{\mathrm{main}} + \epsilon \, p_\theta(\cdot \mid c, x_{<t}), \tag{5}$$

where $q_t^{\mathrm{main}}$ is typically the base distribution sharpened by the *current* proposal exponent $\beta_t$, i.e., $q_t^{\mathrm{main}}(\cdot \mid c, x_{<t}) \propto p_\theta(\cdot \mid c, x_{<t})^{\beta_t}$. The unnormalized importance weight $w_t^{(i)}$ for particle $i$ is updated recursively:

$$w_t^{(i)} \leftarrow w_{t-1}^{(i)} \cdot \frac{p_\theta(x_t^{(i)} \mid c, x_{<t}^{(i)})^{\alpha^\star}}{q_t(x_t^{(i)} \mid c, x_{<t}^{(i)})}. \tag{6}$$

Since the annealing schedule affects only the proposal distribution and the denominator uses the actual mixture probability $q_t$, no additional re-tempering correction is required. The locally sharpened distribution $q_t^{\mathrm{main}}$ is used only as a proposal and is not assumed to be the exact conditional distribution of the global power-shaped target.

## 3.2. Resampling and Population Dynamics

Weight variance grows over time, causing *particle degeneracy*: a single particle may dominate the probability mass (Kong et al., 1994; Doucet et al., 2000b). We track the ESS metric and trigger resampling when $\mathrm{ESS}_t < \tau N$ (typically $\tau = 0.5$). We use a configurable resampling scheme (systematic by default in the main batched implementation), with residual and multinomial resampling supported as alternatives. These standard resampling schemes remove low-weight particles and duplicate high-weight particles while preserving an unbiased particle approximation.

When $\mathrm{ESS}_t \geq \tau N$, the population is sufficiently diverse and we skip resampling entirely: no particles are duplicated,

---

**Algorithm 1** Cache-coherent ASMC for LLM Reasoning

1: **Input:** Prompt $c$, particles $N_{\mathrm{fast}}, N_{\mathrm{hard}}$, block size $B$, thresholds $\tau, \delta_{\mathrm{early}}, \delta_{\mathrm{fast}}$.
2: Set $N \leftarrow N_{\mathrm{fast}}$.
3: **loop**
4:    // Adaptive Pass: Fast → Hard
5:    Initialize $N$ particles and KV cache from prompt $c$.
6:    **for** each token block $k = 1 \ldots T/B$ **do**
7:       **1. Parallel Expand:** Generate $B$ tokens for all particles (Batched).
8:       **2. Weight Update:** Accumulate importance weights over block; compute ESS.
9:       **if** $\mathrm{ESS} < \tau N$ **then**
10:          **3. Resampling:** Select ancestors $a_{1:N}$ using the configured scheme.
11:          **4. Cache-Coherent Reorder:** gather KV caches by $a_{1:N}$ (No Replay).
12:          Reset weights $w \leftarrow 1/N$.
13:       **end if**
14:       **Early Stop Check:** Aggregate normalized weights by parsed answer; if top-answer mass $m_{\mathrm{top}} \geq \delta_{\mathrm{early}}$ and guard conditions hold, **return** top answer.
15:    **end for**
16:    **Adaptive Escalation Check:** Compute weighted vote and top-answer mass $m_{\mathrm{top}}$.
17:    **if** $m_{\mathrm{top}} < \delta_{\mathrm{fast}}$ and $N = N_{\mathrm{fast}}$ **then**
18:       Set $N \leftarrow N_{\mathrm{hard}}$ and **restart** a fresh pass from prompt $c$.
19:    **else**
20:       **break**
21:    **end if**
22: **end loop**
23: **Output:** Weighted vote or top answer.

---

cache reordering is avoided, and weights carry forward unchanged. This keeps system overhead low when the population does not need correction.

## 3.3. Adaptive Compute Allocation

Standard sampling spends the same compute on every problem. ASMC adjusts compute based on difficulty via two mechanisms: *early exit* for easy problems and *population escalation* for hard ones.

**Confidence metric and adaptive policy.** We measure confidence via *top-answer mass*: particles are first mapped to parsed answers, and normalized particle weights are then aggregated by answer. Let $m_{\mathrm{top}} = \max_a \sum_{i:\mathrm{ans}(x^{(i)})=a} \tilde{w} t^{(i)}$ denote the largest mass assigned to any parsed answer $a$. When $m_{\mathrm{top}}$ is high and additional guard conditions are satisfied, the population has strong answer-level agreement and further search offers diminishing returns.

The adaptive logic proceeds as follows (see Algorithm 1): (1) **Early Exit (Easy):** if $m_{\mathrm{top}} \geq \delta_{\mathrm{early}}$ during the Fast Pass ($N_{\mathrm{fast}}$), and the minimum-token, ESS, parsed-fraction, and stability guards are satisfied, we terminate immediately and return the top answer to save compute. (2) **Escalation (Hard):** if the Fast Pass finishes with low answer-level

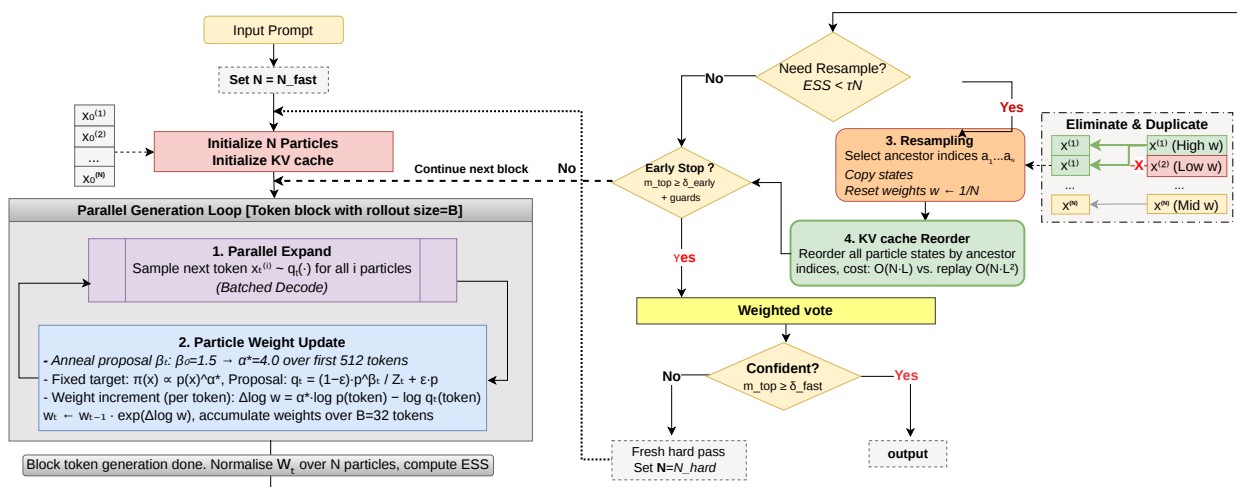

*Figure 1.* **ASMC with Cache Coherence.** ASMC evolves $N$ particles in parallel via batched decoding. Importance reweighting targets the fixed power-shaped distribution $\pi(x) \propto p(x)^{\alpha^\star}$, while an annealed proposal improves exploration and reduces variance. Cache-coherent reorder enables efficient resampling without prefix replay.

confidence ($m_{\text{top}} < \delta_{\text{fast}}$), we discard the fast-pass result and launch a fresh hard pass from the original prompt with a larger population ($N_{\text{hard}}$) to better resolve the multimodal target. (3) **Budget Cap:** we enforce a strict per-instance budget limit $C^\star$ (measured in $C_{\text{int}}$, defined in Section 5). If cumulative attention interactions exceed $C^\star$, inference terminates immediately and returns the current weighted-vote result, ensuring predictable latency bounds.

**Theoretical scope.** The fixed-target SMC core follows standard sequential importance weighting for the specified proposal distribution. In contrast, early exit, fast-to-hard escalation, budget caps, and weighted-vote output are latency-oriented decision rules. They are empirically motivated and are not covered by a formal convergence guarantee for the full adaptive procedure.

## 4. Cache-Coherent Resampling for Transformers

Transformer inference is only fast at test time because it reuses per-layer KV caches. Since our goal is to scale inference with a *population* of particles, we must preserve this KV-cache reuse across particles rather than falling back to repeated prefix recomputation. The difficulty is that resampling in SMC breaks cache locality. After resampling, particle $i$ can no longer keep its own history. It must inherit the prefix of its selected ancestor $a_i$. As a result, every piece of particle-specific state has to be made consistent with the new ancestry, including KV caches at all layers, position indices, attention masks, and any auxiliary tensors used by the decoding loop.

A straightforward implementation quickly runs into a sys-

tems bottleneck. One option is **prefix replay**: drop the caches and recompute the entire inherited prefix by running the model again. This is simple but expensive, because attention over a length-$L$ prefix scales quadratically, so replaying prefixes across $N$ particles costs $O(NL^2)$ attention work and becomes prohibitive for long contexts. The other option is to **copy state**: directly overwrite the state of particle $i$ with that of its ancestor. This avoids recomputation, but it moves a large amount of data, on the order of $O(NLD_{\text{model}})$ elements across layers, and frequent copies can erase the throughput gains of parallel decoding.

**Cache-Coherent State Reordering** We implement resampling as a GPU-friendly primitive by **reordering the particle dimension**. Given the ancestor indices after resampling, we apply the ancestry map as a single gather over the particle axis of the per-layer **KV caches**. Concretely, for each Transformer layer, we gather the KV cache tensors from the selected ancestors into the new particle slots, and we apply the same reordering to all particle-associated tensors, including position IDs, attention masks, and any auxiliary decoding buffers. This realizes resampling without replaying prefixes and without reconstructing caches token by token.

Reordering still moves data proportional to the cached prefix length, but the access pattern is regular and bandwidth-friendly on GPUs. In practice, this is far cheaper and more predictable than recomputing attention over long prefixes. Our microbenchmarks in Section 6.2 show that cache-coherent reordering keeps resampling overhead small even beyond 3000 tokens, which makes frequent resampling feasible at the budgets we study. The complete ASMC procedure is given in Algorithm 1 and illustrated in Figure 1.

# 5. Compute Budgets and Latency-Aware Evaluation

We evaluate test-time scaling using both hardware-agnostic compute proxies and measured wall-clock latency. The same attention interaction count can yield very different runtimes: teacher-forcing/prefill is highly parallel and GPU-efficient, while autoregressive decoding is sequential and often under-utilizes the GPU. A single interaction proxy can therefore be miscalibrated across methods that mix prefill and decode differently. We use **wall-clock p95 latency** as the primary deployment-relevant axis in our figures, and report compute proxies for controlled comparisons and diagnostics.

## 5.1. Compute proxies

We report two proxies computed from a shared forward-pass hook. Let $k$ index forward calls, with active batch size $B_k$ and context length $L_k$.

**Batch-steps.**
$$C_{\text{step}} = \sum_k B_k, \tag{7}$$

where one unit corresponds to one forward call (independent of sequence length). $C_{\text{step}}$ is sensitive to control flow (e.g., resampling frequency, proposal structure) but provides a simple measure of total invocation volume.

**Attention interactions.** We log attention-dominated interaction counts separately for cached decode and prefill/replay:

$$C_{\text{int}} = \underbrace{\sum_{k \in \text{decode}} B_k L_k}_{C_{\text{int}}^{\text{dec}}} + \underbrace{\sum_{k \in \text{prefill/replay}} B_k \frac{L_k(L_k+1)}{2}}_{C_{\text{int}}^{\text{pre}}}. \tag{8}$$

$C_{\text{int}}^{\text{dec}}$ captures the $O(L)$ scaling of cached decoding, while $C_{\text{int}}^{\text{pre}}$ captures the $O(L^2)$ scaling of prefix recomputation (prefill, replay, or rebuild). $C_{\text{int}}$ is *not* an absolute FLOPs measure and does not account for non-attention components (MLP, projections, kernel launch overheads, memory movement). Most importantly, because prefill and decode operate at different effective throughput on modern GPUs, $C_{\text{int}}$ may not align with wall-clock *across methods* that shift work between prefill and decode. We therefore interpret $C_{\text{int}}$ primarily as an *algorithmic interaction budget* and always pair it with latency measurements (p50/p95).

## 5.2. Budget accounting rules

**Per-call accounting.** For each forward call, we compute incremental interaction costs as:

$$\Delta C_{\text{int}} = B \cdot L \qquad \text{(cached decode)} \tag{9}$$

$$\Delta C_{\text{int}} = B \cdot \frac{L(L+1)}{2} \quad \text{(prefill / replay / rebuild)}. \tag{10}$$

Any operation that recomputes attention over a prefix, e.g., teacher-forcing scoring, MCMC truncation proposals with prefix replay, or ASMC full rebuilds, is logged as prefill/replay with quadratic cost. Cached autoregressive decoding steps are logged with linear cost.

**Implementation.** We instrument a shared forward-pass hook across all methods. The hook distinguishes prefill from cached decode via `past_key_values is None`: if true, the call is logged as prefill; otherwise, it is cached decode. For prefill, $L$ is the input sequence length; for decode, $L = \text{past\_len} + 1$. This unified instrumentation ensures consistent logging despite different control flow (batched sampling, particle resampling, or sequential MCMC).

## 5.3. Latency reporting

We measure wall-clock latency end-to-end for each problem instance under fixed hardware and runtime settings, and report **p50** and **p95** across instances. We emphasize p95 because tail behavior dominates user-facing quality-of-service and sharply exposes sequential bottlenecks (e.g., long-hit-limit trajectories in MCMC).

## 5.4. Compute-matched tuning protocol

For controlled compute-matched comparisons, for each target budget $C_{\text{int}}^{\star}$ we perform a limited hyperparameter search (3–5 values per knob) and select the configuration maximizing accuracy subject to a strict cap on *mean* realized compute:

$$\mathbb{E}[C_{\text{int}}] \leq 1.02 \, C_{\text{int}}^{\star}.$$

We then report the corresponding p50/p95 latency for that selected configuration. Full compute-matched results across all budget caps are provided in Appendix Table A1. Main conclusions, however, are drawn from the accuracy–p95 latency frontier, with compute proxies used to interpret where methods spend work (prefill vs decode) rather than as a sole fairness axis.

# 6. Experiments

**Tasks and Metrics.** Our primary benchmark is MATH500, evaluated by exact-match accuracy using robust answer parsing. We additionally report end-to-end wall-clock latency and summarize p50/p95 over problem instances, using **p95** as the primary deployment-facing metric. We enforce a task-appropriate generation cap of $L_{\max} = 3072$ tokens for MATH500. To assess transfer beyond mathematical reasoning while staying within verifiable tasks, we include results on *HumanEval* (code generation) in Appendix E, reported as pass@1 under the same latency reporting protocol. MATH500 is our main focus because it features long-horizon reasoning where tail latency and sequential bottlenecks are most pronounced.

**Models.** MATH500 experiments use *Qwen2.5-Math-7B* (bfloat16), ensuring a controlled comparison under a fixed base model.

**Methods Evaluated.** We compare the following test-time inference procedures under the same base model: **Greedy** decoding ($T=0$) as a deterministic baseline; **Best-of-$n$** (*sample-and-select*), drawing $n$ independent samples and selecting the completion with the highest length-normalized sequence log-probability (details below); **MCMC** (*autoregressive Metropolis–Hastings*, RWS-style), the sequential Markov chain baseline from prior work, implemented with the same model and scoring; **ASMC (fixed-$N$, cache-coherent)**, our particle method with a fixed population size $N$, using cache-coherent resampling (KV reordering) during resampling; **ASMC (adaptive-$N$, cache-coherent)**, our adaptive variant with $N_{\text{easy}} = \frac{1}{2}N$ and $N_{\text{hard}} = N$, also using cache-coherent resampling; and **System ablation (full rebuild)**, ASMC with resampling implemented by rebuilding KV caches (prefix recomputation) instead of cache-coherent reordering. All methods rely only on model log-probabilities $p_\theta(x)$ for generation and for any internal weighting/acceptance decisions.

**Best-of-$n$ baseline.** For best-of-$n$, we draw $n$ independent samples and select

$$\hat{x} = \arg\max_x \frac{1}{|x|} \sum_t \log p_\theta(x_t \mid c, x_{<t}).$$

We account for *all* generation and scoring costs in the compute logs. Candidate generation is batched, and log-probability scoring is implemented via micro-batched teacher-forcing (with adaptive chunk size), so that prefill/scoring contributions are fully reflected in $C_{\text{int}}$ and $C_{\text{step}}$.

**Compute and Latency Accounting.** We log two compute proxies, attention interactions $C_{\text{int}}$ and batch-steps $C_{\text{step}}$ (Section 5), using a shared forward-pass hook across all methods. However, $C_{\text{int}}$ can be *miscalibrated across methods* because different phases have different effective GPU throughput: teacher-forcing/prefill (including scoring) admits high parallelism, whereas autoregressive decoding is inherently sequential. Therefore, we use **wall-clock p95 latency** as the primary comparison axis in our main frontier plots, and use $C_{\text{int}}$ for controlled compute-matched selection and interpretability.

**Compute-matched tuning protocol.** For each target budget cap $C^\star$ (multiples of $C_0$), we perform a limited hyper-parameter search and select the configuration maximizing accuracy subject to $\bar{C}_{\text{int}} \leq 1.02 C^\star$. We then report the corresponding p50/p95 latency for the selected configuration. Full tables and sweep details are provided in the appendix.

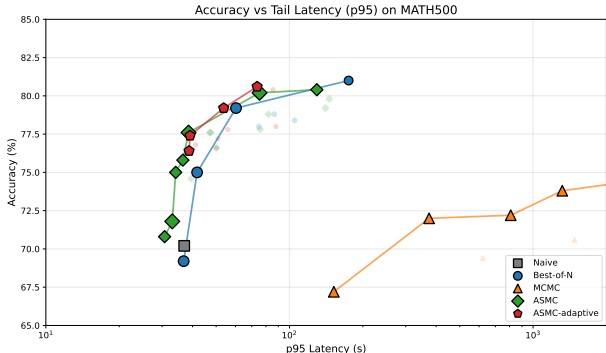

*Figure 2.* **Accuracy–p95 latency frontier on MATH500.** Each marker is the best configuration per method under a compute cap $C^\star$ (multiples of $C_0=0.38$M).

### 6.1. Accuracy–Tail Latency Frontier on MATH500

We evaluate test-time scaling on MATH500 and summarize the **accuracy–tail-latency** trade-off in Figure 2. While we use an interaction budget ($C_{\text{int}}$) to select comparable configurations (Section 5), we emphasize p95 latency as the primary deployment-facing metric, since methods differ substantially in how work is split between prefill (high throughput) and decode (low throughput).

At low-to-moderate budgets, best-of-$n$ improves accuracy quickly but increases tail latency due to waiting for multiple samples: at $8 \times C_0$, best-of-$n$ ($n=2$) reaches 75.0% with p95 41.8s, whereas ASMC ($N=4$) attains 71.8% with a lower p95 of 33.0s. At higher budgets, ASMC trades a small amount of accuracy for substantially lower p95 latency relative to log-probability best-of-$n$, while larger ASMC budgets provide higher-accuracy points beyond best-of-$n$ saturation: at $16 \times C_0$, ASMC ($N=16$) achieves 77.6% with p95 38.5s, compared to best-of-$n$ ($n=4$) at 79.2% with p95 60.3s; the same pattern persists at $32 \times C_0$ (ASMC p95 38.5s vs. best-of-$n$ p95 60.3s with similar accuracy). In contrast, sequential MCMC suffers from extreme tail latency and poor scalability: p95 grows from 152s at $8 \times C_0$ to 1318s at $128 \times C_0$, rendering it impractical despite nontrivial compute expenditure.

Table 1 reports the corresponding compute-matched selections, including $C_{\text{int}}$ and p50/p95 latency. We note that best-of-$n$ saturates at $n=4$ on MATH500 (see Table A1), so larger compute caps do not yield further gains for this baseline. Overall, ASMC and best-of-$n$ form the strongest frontiers. Best-of-$n$ reaches a strong low-latency point but saturates at $n = 4$, whereas ASMC provides higher-accuracy operating points at larger budgets while avoiding the severe tail behavior of sequential MCMC. Appendix F.2 further shows that majority-vote best-of-$n$ can approach ASMC accuracy at large $n$, but with substantially worse p95 latency.

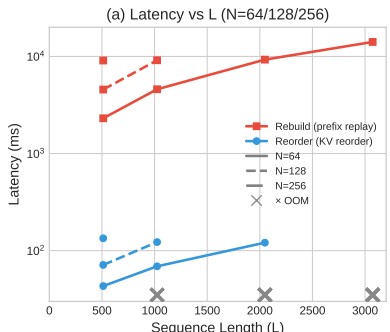
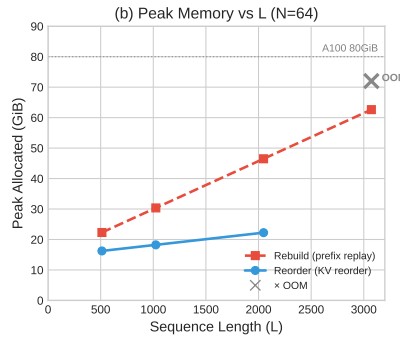
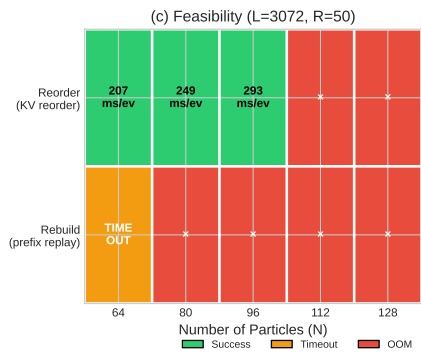

*Figure 3.* **System cost of a resampling event.** We benchmark a single event (resample and resume generation for one token) for Rebuild (prefix replay) and Reorder (KV reorder) on Qwen2.5-Math-7B (bf16). (a) Event latency versus sequence length $L$ (log scale). (b) Peak allocated memory versus $L$ at $N{=}64$. (c) Feasibility at $L{=}3072$ over $R{=}50$ repeated events. $\times$ denotes OOM and TIMEOUT denotes exceeding the per-run threshold. See Appendix A for full hardware specifications.

*Table 1.* **Latency-aware compute-matched results on MATH500.** Best configuration per method under each budget cap $C^*$ (within 2%). Full tables in Appendix Table A1.

| Budget | Method | Acc.% | p50 | p95 |
|---|---|---|---|---|
| 8× | Greedy | 70.2 | 12.0 | 37.0 |
| | Best-of-2 | 75.0 | 12.5 | 41.8 |
| | MCMC ($S{=}2, B{=}8$) | 67.2 | 34.0 | 152.1 |
| | ASMC ($N{=}4$) | 71.8 | 12.5 | 33.0 |
| 16× | Greedy | 70.2 | 12.0 | 37.0 |
| | Best-of-4 | 79.2 | 14.5 | 60.3 |
| | MCMC ($S{=}2, B{=}8$) | 67.2 | 34.0 | 152.1 |
| | ASMC ($N{=}16$) | 77.6 | 14.2 | 38.5 |
| | ASMC-adapt ($N{=}16$) | 76.4 | 13.5 | 38.7 |
| 32× | Greedy | 70.2 | 12.0 | 37.0 |
| | Best-of-4 | 79.2 | 14.5 | 60.3 |
| | MCMC ($S{=}2, B{=}32$) | 72.0 | 58.4 | 374.2 |
| | ASMC ($N{=}16$) | 77.6 | 14.2 | 38.5 |
| | ASMC-adapt ($N{=}32$) | 77.4 | 15.0 | 39.1 |
| 128× | Greedy | 70.2 | 12.0 | 37.0 |
| | Best-of-4 | 79.2 | 14.5 | 60.3 |
| | MCMC ($S{=}8, B{=}32$) | 73.8 | 211.6 | 1317.5 |
| | ASMC ($N{=}64$) | 80.2 | 19.8 | 75.4 |
| | ASMC-adapt ($N{=}128$) | 80.6 | 20.9 | 73.7 |

### 6.2. Systems Microbenchmarks and Boundary Regimes

To assess whether resampling-based inference is practically limited by system overhead, we microbenchmark a single *resampling event* on Qwen2.5-Math-7B (bf16). We compare **full rebuild**, which discards the KV cache and replays prefix prefill, to **KV reorder**, which reindexes the cached KV along the batch dimension and then resumes generation for one token. To avoid conflating the benchmark with full-sequence logits materialization, prefill, replay, and the one-step decode are executed through the model backbone with KV caching enabled.

Figure 3 shows that KV reorder substantially reduces the

system cost of resampling. Across feasible points, reorder improves event latency by **53** to **77**× relative to rebuild (Figure 3a). At $N{=}64$, reorder also uses less event-window peak memory and grows more gently with $L$ than rebuild (Figure 3b). In the boundary regime at $L{=}3072$ with $R{=}50$ repeated events, rebuild becomes infeasible at $N{\geq}80$ due to OOM and consistently exceeds the per-run latency budget even at $N{=}64$, whereas reorder remains feasible through $N{=}96$ at approximately 207 to 293 ms/event before hitting device limits at $N{\geq}112$ (Figure 3c). Together, these results indicate that cache-coherent resampling is a necessary system enabler for the high-budget settings studied in Section 6.1.

### 6.3. Particle Collapse Analysis and Mitigation

We study *particle collapse* as a systems- and algorithm-level failure mode of ASMC on MATH500. We instrument ASMC to record, per problem, (i) the number of resampling events and (ii) a normalized collapse diagnostic $\mathrm{ESS}_{\min}/N$. Fig. 4(a) shows a strong monotonic degradation as resampling becomes frequent: accuracy drops from 100% for problems with 1-2 resampling events to 43.9% for those with $\geq 16$ events. Thus, repeated resampling is not merely additional compute, but strongly predictive of failure.

Fig. 4(b) connects this behavior to particle impoverishment. Incorrect problems exhibit substantially lower $\mathrm{ESS}_{\min}/N$ than correct ones ($p < 10^{-15}$, Mann–Whitney), confirming that collapse correlates with errors. Using $\mathrm{ESS}_{\min}/N \leq 0.1$ as a diagnostic threshold defines a collapse-prone "hard" subset covering 70% of problems, which we use for targeted analysis.

Finally, we test whether the *resampling scheme* itself materially affects outcomes by comparing residual resampling (low-variance, partially deterministic) against multinomial resampling under paired, problem-level evaluation with ma-

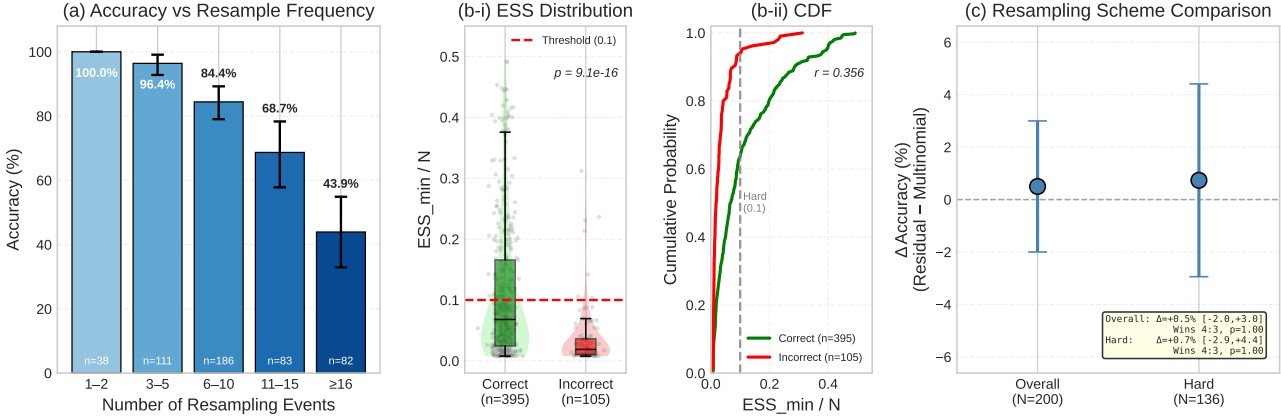

*Figure 4.* **Diagnosing particle collapse.** (a) Accuracy vs. number of resampling events. (b) $\mathrm{ESS}_{\min}/N$ distribution for correct vs. incorrect problems. (c) Residual vs. multinomial resampling (paired, 200 problems, majority vote over 3 seeds).

jority vote over 3 random seeds (Fig. 4(c)). On the matched set ($N{=}200$), residual yields only a small, statistically non-significant improvement both overall ($\Delta{=}{+}0.5\%$) and on the hard subset ($\Delta{=}{+}0.7\%$). Taken together, these results suggest that *collapse severity and resampling frequency are the dominant drivers of ASMC failures in our setting*, while switching between standard resampling schemes has a smaller second-order effect. We additionally sweep the ESS threshold in Appendix F.3; performance varies only slightly across $\tau \in \{0.3, 0.5, 0.7\}$, suggesting limited sensitivity in our current setting.

## 7. Related Works

**Test-Time Scaling and Power-Shaped Distributions.** Test-time scaling improves reasoning performance without additional training, typically via independent sampling strategies like *best-of-n* or *self-consistency* (Wei et al., 2022; Wang et al., 2023). However, these approaches allocate compute uniformly and do not explicitly target structured trajectory distributions. Recent work argues for sampling from *power-shaped* distributions $\pi(x) \propto p_\theta(x)^\alpha$ to concentrate probability mass on high-quality reasoning traces (Karan & Du, 2025), drawing on concepts from annealed importance sampling and likelihood tempering (Neal, 2001; Friel & Pettitt, 2008). By amplifying the relative probability of coherent paths, this formulation reveals reasoning capabilities that temperature scaling alone cannot achieve. While Metropolis-Hastings (MH) has been used to target these distributions (Karan & Du, 2025), it is inherently sequential and mixes slowly over long trajectories. The serial nature of chain-based MCMC "generate-then-verify" loops wastes compute on rejections and underutilizes the massive parallelism of modern GPUs. This mismatch motivates particle-based inference, which approximates sharpened distributions efficiently at high budgets through parallelism.

**Sequential Monte Carlo and Adaptive Inference.** Sequential Monte Carlo (SMC) approximates target distributions via a parallel population of weighted particles, offering a flexible alternative to chain-based MCMC (Doucet et al., 2000a; Chopin & Papaspiliopoulos, 2020). Despite its success in state-space models and robotics (Thrun et al., 2005), SMC is under-utilized in LLMs due to the challenges of *particle degeneracy* and the prohibitive cost of managing Transformer KV caches during resampling. Standard resampling schemes can mitigate weight degeneracy (Liu & Chen, 1998), but their application to long-horizon autoregressive generation requires careful system-level integration to avoid memory bottlenecks. We revisit SMC for LLM reasoning by introducing *cache-coherent resampling* to eliminate prefix recomputation overhead, making population-based inference computationally viable. Beyond classical SMC diagnostics, we show empirically that resampling frequency predicts decoding failure in LLMs. To improve the accuracy-compute trade-off, we introduce top-answer-mass early exit and hardness-dependent particle allocation, which adjust compute based on problem difficulty.

## 8. Discussion

We identify a systems bottleneck in sequential trajectory sampling for test-time scaling. Prior work shows that chain-based MCMC targeting power-shaped trajectory distributions can be algorithmically strong and can approach GRPO-style performance without additional training. However, its serial control flow and decode-heavy execution lead to heavy tail latency at high budgets, which dominates deployment cost. ASMC addresses this mismatch by replacing a single chain with batched particles and by providing scaling knobs that align with GPU parallelism.

A second barrier is cache management. Resampling changes particle ancestry and naively requires prefix replay to rebuild

KV caches. Cache-coherent resampling resolves this by reordering KV caches under the ancestor mapping, which turns particle resampling into a practical primitive for Transformer decoding and enables scaling to longer contexts and larger particle counts.

We emphasize deployment-facing evaluation using p95 latency. Interaction-based accounting supports controlled compute-matched selection and interpretability, but cross-method comparisons can be miscalibrated when methods shift work between high-throughput prefill or teacher-forcing and low-throughput autoregressive decoding. Reporting both latency and accounting signals clarifies where each method spends computation and why tail behavior differs. In addition, best-of-$n$ improves quickly at small budgets but saturates on MATH500 and exhibits higher tail latency as $n$ increases because the runtime is gated by the slowest sample.

We also observe that particle collapse is a key failure mode. Low $\mathrm{ESS}_{\min}/N$ strongly predicts errors, while the specific resampling scheme has limited effect, which suggests that maintaining diversity and avoiding degeneracy is more important than fine-grained resampling choices. We include additional results on HumanEval in Appendix E to probe transfer beyond mathematical reasoning within verifiable tasks.

**Limitations and future work.** Our conclusions are based on a fixed model and runtime setting, and the relative throughput of prefill and decode can vary with hardware, kernels, and serving concurrency. We have not benchmarked 70B+ or multi-device settings. The cost of cache-coherent reordering is dominated by KV-cache data movement, with per-event traffic scaling roughly as

$$O(2NL\, n_{\text{layers}}\, n_{\text{kv heads}}\, d_{\text{head}})$$

elements, where the factor 2 accounts for keys and values, up to read/write factors. The crossover point between gather-based reordering and prefix replay may depend on cache layout, GQA/MQA, paged-attention implementation, kernel support, and inter-device communication. Future work includes stronger rejuvenation moves to improve diversity, more adaptive policies that allocate particles based on online degeneracy signals, and extending cache-coherent primitives to additional inference backends and architectures.

## Acknowledgements

This work was supported by the United Kingdom Research and Innovation (grant EP/S02431X/1), UKRI Centre for Doctoral Training in Biomedical AI at the University of Edinburgh, School of Informatics.

## Impact Statement

Our approach improves inference-time reasoning by enabling more efficient test-time scaling for large language models. Potential positive impacts include higher reliability in applications such as education, scientific assistance, and decision support, where better reasoning quality can reduce user burden and error rates. A key consideration is that test-time scaling increases compute and energy consumption relative to standard decoding, which can raise operational cost and environmental footprint. Our cache-coherent resampling mechanism reduces wasted recomputation and can improve hardware efficiency for a given accuracy target, but it does not eliminate the cost increase from using larger inference budgets. We encourage practitioners to report deployment-relevant metrics, including tail latency and resource usage, and to apply test-time scaling in proportion to task risk, with appropriate monitoring and governance.

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

# Appendix Overview

This appendix is organized as follows:

- Appendix A: Details the experimental environment and hardware.

- Appendix B: Presents the sanity check validating our compute proxy $C_{\mathrm{int}}$.

- Appendix C: Provides budget accounting rules and full compute-matched results (Table A1).

- Appendix D: Covers implementation details for baselines, resampling, and algorithmic comparisons.

- Appendix E: HumanEval supplementary results.

- Appendix F: Provides additional ablations and dataset results.

- Appendix G: Offers an extended review of probabilistic inference methods (SMC, MCMC, and Filtering).

## A. Experimental Environment

All experiments and microbenchmarks were conducted on compute nodes equipped with NVIDIA A100-SXM-80GB GPUs. We used *Qwen2.5-Math-7B* and *Qwen2.5-7B* models loaded in **bfloat16** precision for all inference runs. The environment uses PyTorch with CUDA 12.x support. For latency measurements, we utilized CUDA events with explicit synchronization to ensure accurate timing of GPU kernels, independent of CPU-side dispatch overhead.

## B. Sanity Check: $C_{\mathrm{int}}$ vs Wall-Clock Latency

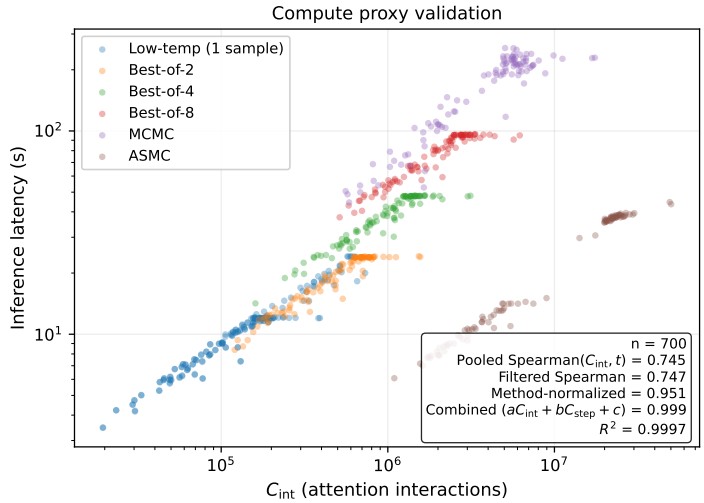

*Figure A1.* **Validation of compute accounting (and its limits).** Scatter plot of attention-interaction budget $C_{\mathrm{int}}$ versus wall-clock inference latency over 700 runs on 100 MATH problems across multiple inference procedures (naive sampling, best-of-$n$ with $n \in \{2, 4, 8\}$, ASMC, and MCMC). Across all methods, $C_{\mathrm{int}}$ exhibits a significant monotonic correlation with latency (Spearman $\rho = 0.745$, $p = 4.3 \times 10^{-125}$), spanning a $2.6 \times 10^3$ dynamic range in $C_{\mathrm{int}}$ (from $1.95 \times 10^4$ to $5.02 \times 10^7$). However, the pooled correlation is dampened by method-dependent overheads and, importantly, by differences in *execution mode*: teacher-forcing/prefill admits high parallelism and high GPU utilization, whereas autoregressive decoding is inherently sequential. After normalizing out method-specific scale factors, the correlation strengthens (method-normalized Spearman $\rho = 0.951$), indicating that $C_{\mathrm{int}}$ reliably captures relative interaction growth *within* a method. Furthermore, augmenting $C_{\mathrm{int}}$ with batch-steps $C_{\mathrm{step}}$ yields an almost perfect latency predictor $\hat{t} = aC_{\mathrm{int}} + bC_{\mathrm{step}} + c$ (Spearman $\rho = 0.999$, $R^2 = 0.9997$), with the largest gains observed for MCMC, consistent with additional control-flow and kernel overheads. Overall, $C_{\mathrm{int}}$ provides a useful, hardware-agnostic accounting signal for compute-matched tables and diagnostics, but we draw our main conclusions using wall-clock latency (p50/p95) to avoid cross-method miscalibration when workloads mix prefill and decode differently.

**Correlation Analysis.** When pooling across inference procedures, wall-clock latency depends not only on attention interactions but also on method-specific control flow, kernel-launch overheads, and the effective throughput of different

phases. In particular, prefill/teacher-forcing can achieve much higher GPU utilization than autoregressive decoding, so the same interaction count may translate to different wall-clock time across methods. This explains why the pooled Spearman correlation between $C_{int}$ and latency is moderate ($\rho = 0.745$), despite consistently strong within-method correlations (e.g., $\rho \in [0.825, 0.987]$ across methods). After removing method-specific scale factors, the correlation becomes strong (method-normalized $\rho = 0.951$), indicating that $C_{int}$ captures relative compute variation beyond constant per-method overheads. Consistently, a simple linear model $\hat{t} = aC_{int} + bC_{step} + c$ nearly perfectly predicts latency ($\rho = 0.999$, $R^2 = 0.9997$), highlighting the role of batch-steps $C_{step}$ for control-flow-heavy methods such as MCMC. For this reason, we report both $C_{int}$ and latency throughout, using $C_{int}$ for controlled compute-matched selection and latency (especially p95) as the primary deployment-facing comparison.

## C. Budget Accounting and Compute-Matched Tuning

**Budget Accounting Rules.** Any operation that recomputes attention over a prefix (proposal replay, cache rebuild) is logged as prefill/replay with quadratic cost; cached one-token decode is logged with linear cost. Our $C_{int}$ proxy measures per-sample attention workload aggregated over forward calls. In batched execution, multiple samples share a single forward call, changing the mapping between $C_{int}$ and wall-clock latency due to hardware parallelism. We therefore use $C_{int}$ for compute-matched selection and report wall-clock p50/p95 latency as the primary deployment metric.

Table A1 provides the full compute-matched comparison on MATH500 across all budget caps ($2\times$–$128\times C_0$).

*Table A1.* **Full compute-matched comparison on MATH500.** For each budget cap $C^*$ (multiples of $C_0$, the naive single-sample cost), we select the best configuration per method under $\bar{C}_{int} \leq 1.02C^*$. $C_0 = 0.38M$.

| Budget | Method | Hyperparams | #Eval | Acc.% (95% CI) | $C_{int}$ (M) | $C_{call}$ | p50(s) | p95(s) |
|---|---|---|---|---|---|---|---|---|
| $2xC_0$ | naive | – | 500 | 70.2 [66.2, 74.2] | 0.38 | 628 | 12.0 | 37.0 |
| | bestofn | $n=1$ | 500 | 69.2 [65.2, 73.2] | 0.76 | 628 | 12.0 | 36.8 |
| $4xC_0$ | naive | – | 500 | 70.2 [66.2, 74.2] | 0.38 | 628 | 12.0 | 37.0 |
| | bestofn | $n=1$ | 500 | 69.2 [65.2, 73.2] | 0.76 | 628 | 12.0 | 36.8 |
| | asmc | $N=4, T_{ann}=512$ | 500 | 71.8 [67.8, 75.6] | 1.31 | 611 | 12.5 | 33.0 |
| $8xC_0$ | naive | – | 500 | 70.2 [66.2, 74.2] | 0.38 | 628 | 12.0 | 37.0 |
| | bestofn | $n=2$ | 500 | 75.0 [71.0, 78.8] | 1.99 | 720 | 12.5 | 41.8 |
| | mcmc | $S=2, B=8$ | 500 | 67.2 [63.2, 71.2] | 2.72 | 2150 | 34.0 | 152.1 |
| | asmc | $N=4, T_{ann}=512$ | 500 | 71.8 [67.8, 75.6] | 1.31 | 611 | 12.5 | 33.0 |
| $16xC_0$ | naive | – | 500 | 70.2 [66.2, 74.2] | 0.38 | 628 | 12.0 | 37.0 |
| | bestofn | $n=4$ | 500 | 79.2 [75.6, 82.6] | 5.27 | 827 | 14.5 | 60.3 |
| | mcmc | $S=2, B=8$ | 500 | 67.2 [63.2, 71.2] | 2.72 | 2150 | 34.0 | 152.1 |
| | asmc | $N=16, T_{ann}=512$ | 500 | 77.6 [74.0, 81.2] | 5.92 | 670 | 14.2 | 38.5 |
| | asmc-adaptive | $N=16, T_{ann}=512$ | 500 | 76.4 [72.6, 80.0] | 3.83 | 696 | 13.5 | 38.7 |
| $32xC_0$ | naive | – | 500 | 70.2 [66.2, 74.2] | 0.38 | 628 | 12.0 | 37.0 |
| | bestofn | $n=4$ | 500 | 79.2 [75.6, 82.6] | 5.27 | 827 | 14.5 | 60.3 |
| | mcmc | $S=2, B=32$ | 500 | 72.0 [68.0, 75.8] | 8.65 | 4853 | 58.4 | 374.2 |
| | asmc | $N=16, T_{ann}=512$ | 500 | 77.6 [74.0, 81.2] | 5.92 | 670 | 14.2 | 38.5 |
| | asmc-adaptive | $N=32, T_{ann}=512$ | 500 | 77.4 [73.6, 81.0] | 8.90 | 734 | 15.0 | 39.1 |
| $64xC_0$ | naive | – | 500 | 70.2 [66.2, 74.2] | 0.38 | 628 | 12.0 | 37.0 |
| | bestofn | $n=4$ | 500 | 79.2 [75.6, 82.6] | 5.27 | 827 | 14.5 | 60.3 |
| | mcmc | $S=8, B=16$ | 500 | 72.2 [68.2, 76.0] | 14.73 | 11175 | 150.3 | 809.6 |
| | asmc | $N=16, T_{ann}=512$ | 500 | 77.6 [74.0, 81.2] | 5.92 | 670 | 14.2 | 38.5 |
| | asmc-adaptive | $N=64, T_{ann}=1536$ | 500 | 79.2 [75.6, 82.8] | 17.56 | 738 | 16.2 | 53.7 |
| $128xC_0$ | naive | – | 500 | 70.2 [66.2, 74.2] | 0.38 | 628 | 12.0 | 37.0 |
| | bestofn | $n=4$ | 500 | 79.2 [75.6, 82.6] | 5.27 | 827 | 14.5 | 60.3 |
| | mcmc | $S=8, B=32$ | 500 | 73.8 [70.0, 77.6] | 25.91 | 18367 | 211.6 | 1317.5 |
| | asmc | $N=64, T_{ann}=1024$ | 500 | 80.2 [76.6, 83.6] | 26.91 | 712 | 19.8 | 75.4 |
| | asmc-adaptive | $N=128, T_{ann}=1536$ | 500 | 80.6 [77.0, 84.0] | 32.67 | 736 | 20.9 | 73.7 |

# D. Implementation Details

## D.1. Resampling Implementation

The main batched implementation uses systematic resampling by default. Residual and multinomial resampling are also supported and are used in the resampling-scheme comparison. For residual resampling, we include numerical stability checks (e.g., handling floating-point errors in weight summation) and fallbacks to multinomial resampling if residual counts are inconsistent, though this is rare in practice (Liu & Chen, 1998; Douc et al., 2005).

**Protocol for Resampling Scheme Comparison.** We evaluate residual vs. multinomial using paired, problem-level aggregation across three seeds (majority vote). We report paired bootstrap CIs for accuracy differences and McNemar's test on the aggregated correctness indicators. Earlier seed-level pooling over (problem, seed) pairs can overstate significance because observations are not independent; we therefore use problem-level aggregation as the primary analysis. The resulting paired test has limited power here (7 discordant problems), so we interpret Fig. 4c as evidence that scheme choice is a second-order factor under our current settings.

## D.2. Baseline Implementation

**Best-of-$n$ (Sample-and-Select).** We draw $n$ independent samples from the base model under the same decoding constraints. We select the final output by maximizing length-normalized sequence log-probability, $\arg\max_x \frac{1}{|x|} \sum_t \log p_\theta(x_t \mid c, x_{<t})$. Best-of-$n$ uses batched generation with micro-batching when GPU memory is limited. Scoring uses teacher-forcing with micro-batched forward passes: when a batch OOMs, we automatically halve the scoring chunk size (e.g., $64 \to 32 \to 16 \to \cdots$) until the forward pass succeeds. All prefill operations are fully accounted in $C_{\text{int}}$: the total prefill count is $\lceil n/\texttt{gen\_chunk}\rceil + \lceil n/\texttt{score\_chunk}\rceil$.

**MCMC (Power MH).** We implement block-wise truncation proposals. For each step, we report block size, proposal temperature, and step counts. Figure A2 illustrates the structural difference between the sequential MCMC (MH) approach and our parallel ASMC approach.

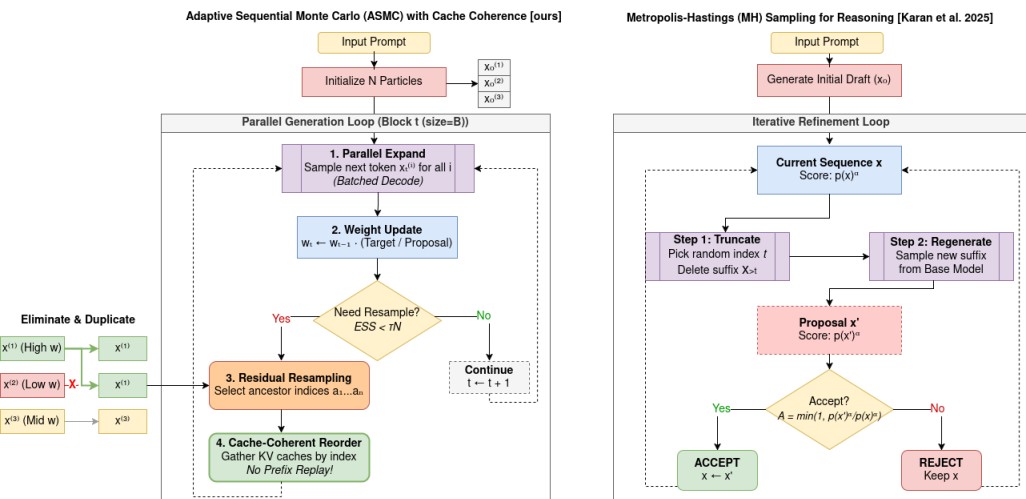

*Figure A2.* **Algorithmic comparison of ASMC (Ours) vs. MH (Karan & Du, 2025). (Right)** MH operates sequentially: it mutates a single sequence $x$ via truncation and regeneration, then accepts or rejects the change. This serial dependency prevents batched execution, and rejected steps waste compute. **(Left)** ASMC evolves a population of $N$ particles in parallel. All particles are extended simultaneously using batched decoding. Instead of rejection, ASMC uses **Reweighting** and **Resampling** to guide the population toward the target distribution $\pi(x) \propto p(x)^\alpha$. Importantly, the **Cache-Coherent Reorder** step (System) enables efficient resampling without the prohibitive cost of prefix replay required in standard schemes.

# E. HumanEval Supplementary Results

**E6: HumanEval setup (supplementary).** We evaluate on HumanEval (164 problems) using Qwen2.5-Math-7B in bfloat16 with a maximum generation length of $L_{\max} = 3072$ and a fixed random seed (0). Unless noted otherwise, sampling

*Table A2.* **Supplementary results on HumanEval (164 problems).** We report pass@1 with 95% CI, along with p50/p95 end-to-end latency (seconds). $C_0 = 0.83M$ denotes the mean $C_{int}$ of greedy decoding.

| Budget | Method | pass@1% [95% CI] | $\bar{C}_{int}$ | p50(s) | p95(s) |
|---|---|---|---|---|---|
| *Baseline:* | | | | | |
| – | Greedy | 55.5 [47.9, 63.1] | 0.83 | 4.1 | 73.0 |
| $8 \times C_0$ | Sampling ($n$=1) | 55.5 [47.9, 63.1] | 1.65 | 4.1 | 73.1 |
| | ASMC ($N$=4) | 49.4 [41.8, 57.0] | 6.92 | 21.4 | 133.4 |
| $16 \times C_0$ | Best-of-$n$ ($n$=4) | 56.1 [48.5, 63.7] | 12.32 | 6.3 | 75.2 |
| | ASMC ($N$=4) | 49.4 [41.8, 57.0] | 6.92 | 21.4 | 133.4 |
| | MCMC ($S$=2, $B$=8) | 51.8 [44.2, 59.5] | 8.02 | 10.6 | 388.3 |
| $32 \times C_0$ | Best-of-$n$ ($n$=4) | 56.1 [48.5, 63.7] | 12.32 | 6.3 | 75.2 |
| | ASMC ($N$=8) | 60.4 [52.7, 67.5] | 14.03 | 18.1 | 155.4 |
| | MCMC ($S$=2, $B$=8) | 51.8 [44.2, 59.5] | 8.02 | 10.6 | 388.3 |
| $64 \times C_0$ | Best-of-$n$ ($n$=4) | 56.1 [48.5, 63.7] | 12.32 | 6.3 | 75.2 |
| | ASMC ($N$=16) | 57.9 [50.3, 65.2] | 32.86 | 20.1 | 173.5 |
| | MCMC ($S$=2, $B$=8) | 51.8 [44.2, 59.5] | 8.02 | 10.6 | 388.3 |
| $128 \times C_0$ | Best-of-$n$ ($n$=4) | 56.1 [48.5, 63.7] | 12.32 | 6.3 | 75.2 |
| | ASMC ($N$=64) | 64.0 [56.7, 71.4] | 88.84 | 9.8 | 200.3 |
| | MCMC ($S$=16, $B$=16) | 56.1 [48.5, 63.7] | 106.04 | 116.3 | 4873.3 |

uses temperature $T = 0.25$. We report pass@1 together with end-to-end latency (p50/p95) and the attention-interaction accounting signal $C_{int}$, where $C_0$ denotes the mean $C_{int}$ of greedy decoding ($C_0 = 0.83M$).

**Methods.** Greedy uses deterministic decoding ($T = 0$). Sampling ($n = 1$) uses a single stochastic sample at $T = 0.25$. Best-of-$n$ generates $n$ samples at $T = 0.25$ and selects the completion with the highest length-normalized sequence log-probability. MCMC is an autoregressive Metropolis–Hastings baseline with block proposals. ASMC is our annealed SMC inference with resampling and weighted voting, using a fixed particle count $N$ and no adaptive easy-hard schedule in this experiment.

**ASMC hyperparameters.** We use anneal_tokens $= 512$, block_size $= 32$, ESS threshold $= 0.5$, vote_mode `weighted`, and systematic resampling.

**Summary.** Best-of-$n$ with log-prob reranking saturates quickly: from single-sample to $n$=4 yields only marginal gains ($\sim$55.5% $\rightarrow$ 56.1%), while p95 remains around 75s. Second, ASMC can substantially improve pass@1 at high budgets, reaching 64.0% at $128 \times C_0$, but at the cost of higher tail latency (p95 $\sim$200s), indicating that high-budget gains on code tasks exist but are "slower". Finally, MCMC exhibits extreme tail latency, with p95 reaching 4873 at $128 \times C_0$, rendering it nearly undeployable for this task. Overall, HumanEval further supports our core system conclusion: chain-based methods carry severe tail latency risk, while particle-based methods scale better, though the optimal accuracy-latency trade-off point varies across tasks.

# F. Additional Ablations and Dataset Results

### F.1. GSM8K Compute-Matched Evaluation

Table A3 reports compute-matched GSM8K results under the same latency protocol as the main text. ASMC substantially reduces p95 latency relative to sequential MCMC while improving accuracy.

### F.2. Majority-Vote Best-of-$n$ on MATH500

Table A4 evaluates majority-vote best-of-$n$ on MATH500. High-$n$ best-of-$n$ can approach ASMC accuracy, but incurs substantially worse tail latency.

Our base-model-only log-probability reranking interface is weaker than verifier- or PRM-based best-of-$n$, and should not be interpreted as representing those stronger reranking pipelines.

*Table A3.* Compute-matched results on GSM8K.

| Budget | Method | Accuracy (%) | p95 Latency (s) |
|---|---|---|---|
| $8 \times C_0$ | MCMC | 84.9 | 151.9 |
| $8 \times C_0$ | ASMC | 90.8 | 13.5 |
| $32 \times C_0$ | MCMC | 85.5 | 413.8 |
| $32 \times C_0$ | ASMC | 90.5 | 16.5 |

*Table A4.* Majority-vote best-of-$n$ on MATH500.

| Method | Accuracy (%) | p95 Latency (s) |
|---|---|---|
| Best-of-4, majority vote | 75.8 | – |
| Best-of-8, majority vote | 79.2 | – |
| Best-of-16, majority vote | 80.0 | 916.0 |

### F.3. ESS Threshold Sensitivity

Table A5 shows that ASMC is not overly sensitive to the ESS threshold in the current setting.

*Table A5.* ESS threshold sensitivity on MATH500.

| ESS threshold $\tau$ | Accuracy (%) |
|---|---|
| 0.3 | 79.4 |
| 0.5 | 79.2 |
| 0.7 | 79.0 |

The adaptive policy, including early exit and fast-to-hard escalation, remains an empirical heuristic rather than a formal convergence guarantee.

## G. Extended Related Work

### G.1. Test-Time Scaling for LLM Reasoning

Test-time scaling has emerged as an effective way to improve reasoning performance of large language models without additional training (Wei et al., 2022; Kojima et al., 2022). A widely studied class of approaches relies on independent sampling with aggregation, including *best-of-n* decoding and *self-consistency* methods that select or vote among multiple independently generated solutions (Kojima et al., 2022; Wang et al., 2023). Related techniques apply re-ranking based on model likelihoods or heuristic scoring functions, scaling compute by increasing the number of samples (Cobbe et al., 2021; Uesato et al., 2022). While effective, these approaches treat samples independently and do not explicitly target a structured distribution over reasoning trajectories. As a result, compute is allocated uniformly across independently generated full trajectories, without adaptive reallocation during generation. In contrast, our work targets trajectory-level test-time inference, where compute is dynamically reallocated based on intermediate evidence via particle reweighting and resampling.

### G.2. Power Distribution and Trajectory-Level Sampling

The use of power-shaped distributions to concentrate probability mass on high-density modes has a rich history in statistical physics (Neal, 2001) and has recently driven significant improvements in continuous generative modeling. Specifically, targeting sharpened distributions ($p^\alpha$) has been shown to enhance sample quality in diffusion and flow models (Skreta et al., 2025; Xu et al., 2025) and has been effective in generating valid structures for protein design (Geffner et al., 2025). In the context of large language models (LLMs), recent work (Karan & Du, 2025) argues that similar gains in reasoning performance can be obtained by sampling from power-shaped trajectory distributions of the form $\pi(x \mid c) \propto p_\theta(x \mid c)^\alpha$ with $\alpha > 1$. This formulation is closely related to classical ideas in probabilistic modeling, including power posteriors and likelihood tempering in Bayesian inference (Friel & Pettitt, 2008), Annealed Importance Sampling (Neal, 2001), and reweighted objectives like Reweighted Wake-Sleep (Bornschein & Bengio, 2014). Within NLP, similar Monte Carlo

mechanics have notably been adapted to target quality-aware posterior distributions for machine translation (Faria et al., 2024). Importantly, this trajectory-level power shaping is distinct from standard token-wise temperature scaling, as the latter introduces stepwise normalization effects that do not correspond to a global power transformation over full sequences.

At inference time, Karan et al. (Karan & Du, 2025) used Metropolis-Hastings, an MCMC sampling method, to sample from the target sharpened power distribution $p_\theta(x \mid c)^\alpha$, and achieved reasoning performance comparable to state-of-the-art RL methods (e.g., GRPO) without any additional training while avoiding the mode collapse and loss of diversity, demonstrating that latent reasoning abilities can be unlocked via inference-time compute alone. However, MCMC methods are inherently sequential, mix slowly over long trajectories, and become increasingly inefficient in high-budget regimes, which motivates alternative inference paradigms that better align with modern parallel hardware.

### G.3. Sequential Monte Carlo and Particle-Based Inference

Sequential Monte Carlo (SMC) methods (Doucet et al., 2000a; Chopin & Papaspiliopoulos, 2020), also known as particle filters (Crisan, 2001; Gordon et al., 1993), approximate target distributions using weighted particle populations with recursive resampling to control weight degeneracy (Liu & Chen, 1998; Douc et al., 2005). SMC and other particle-based inference methods (Huang, 2024) have been widely applied in state-space models (Doucet et al., 2001; Hargreaves et al., 2025), robotics (Thrun et al., 2005), and reinforcement learning (Bryce et al., 2006; Piché et al., 2019), where they provide a flexible alternative to chain-based MCMC (more details about MC methods can be found in Appendix.G.5). Despite their maturity, SMC methods have seen limited adoption for long-horizon autoregressive text generation, particularly in LLMs. Existing applications typically assume short sequences or problem structures that do not require tight integration with Transformer decoding mechanics (Lew et al., 2023; Zhao et al., 2024). Our work revisits SMC in this setting, showing that with appropriate system level support it can serve as an effective test-time inference primitive for LLM reasoning.

### G.4. Particle Degeneracy and Adaptive Inference Strategies

Particle degeneracy and collapse are well-known issues in SMC, commonly monitored via *effective sample size* (ESS) and mitigated through alternative resampling schemes such as residual or stratified resampling (Liu & Chen, 1998; Douc et al., 2005; Huang, 2025). While these techniques are well understood in classical settings, their behavior in long-horizon, high-variance autoregressive reasoning tasks has not been systematically studied.

We complement prior SMC diagnostics by quantifying how collapse metrics correlate with LLM decoding failures, and by ablating the resampling scheme under paired, problem-level aggregation. Our analysis shows that resampling frequency is a stronger predictor of failure than the choice of resampling scheme. Furthermore, to address the computational overhead of population-based methods, we introduce two adaptive strategies: *adaptive cost control*, which utilizes top-mass concentration for early exits, and *hardness-adaptivity*, which dynamically escalates particle allocation for difficult problem instances.

### G.5. Probabilistic Inference Methods

Various approaches can be used for estimating posterior distributions, each with distinct trade-offs regarding computational efficiency, scalability, and handling of sequential data. Classic Monte Carlo methods, such as Importance Sampling (IS), provide unbiased estimates under mild conditions (Kloek & van Dijk, 1978) and are straightforward to implement; however, they suffer from the curse of dimensionality. In high-dimensional spaces, the variance of importance weights typically grows exponentially with dimension (often referred to as *weight degeneracy*), rendering IS impractical for complex models and long horizons (van Leeuwen, 2003; Bengtsson et al., 2008).

Markov Chain Monte Carlo (MCMC) methods, including Metropolis-Hastings (MH (Metropolis et al., 1953)) and Hamiltonian Monte Carlo (HMC (Duane et al., 1987)), overcome some of these dimensional limitations by constructing a Markov chain that converges to the target distribution. While asymptotically exact and powerful for static distributions, MCMC methods are inherently serial and computationally expensive for sequential tasks; updating the posterior with new data often requires either re-running the chain or employing specialized schemes, making them ill-suited for real-time or stepwise reasoning tasks.

Stein Variational Gradient Descent (SVGD (Liu & Wang, 2016)) provides a deterministic, particle-based alternative that casts sampling as an optimization problem. While SVGD is sample-efficient, it can suffer from particle collapse and systematic underestimation of posterior variance, particularly in multimodal targets.

In the realm of dynamic state estimation, the Kalman Filter (KF) remains the optimal Bayesian estimator for linear dynamical systems (Kalman, 1960). Extensions such as the Extended (EKF) and Unscented (UKF) Kalman Filters (Julier & Uhlmann, 2004) address nonlinearity but still approximate the posterior as Gaussian, struggling to capture the discrete, branching structures of linguistic reasoning.

Sequential Monte Carlo (SMC (Doucet et al., 2000a)), or particle filtering, addresses many of these limitations. Unlike MCMC, SMC is explicitly designed for online inference; unlike the KF, it makes no Gaussian assumptions and can represent arbitrary multimodal distributions (Gordon et al., 1993). This capability is critical for reasoning tasks involving high ambiguity and combinatorial state spaces.

### The Bootstrap Filter

The bootstrap particle filter (Gordon et al., 1993), also known as Sampling-Importance-Resampling (SIR), represents the filtering distribution using a weighted set of random samples. It operates recursively via prediction, update, and resampling stages. This mechanism mitigates degeneracy by eliminating low-weight particles and replicating high-weight ones, forming the basis of our ASMC implementation.

### The Extended Kalman Filter (EKF)

The EKF approximates nonlinear systems via local linearization (Taylor expansion) (Jazwinski, 1970). While computationally efficient, it maintains a unimodal Gaussian approximation and can diverge in highly nonlinear or multimodal regimes, making it poorly suited for the discrete branching nature of language generation.

### The Ensemble Kalman Filter (EnKF)

The EnKF (Evensen, 1994) represents uncertainty using an ensemble but updates them using a Monte Carlo implementation of the Kalman gain rather than likelihood reweighting. While effective in high-dimensional physical systems (e.g., meteorology), its reliance on second-order statistics (mean and covariance) makes it prone to failure in the strongly non-Gaussian posteriors typical of LLM reasoning.

