# OpenReview forum: "Cache Coherent Resampling for Efficient Test Time Scaling in LLM Reasoning via Adaptive Sequential Monte Carlo"
_ICML.cc/2026/Conference — ICML 2026 regular_

### Official Review · Reviewer_9ckj · 2026-03-08

**Soundness:** 3
**Presentation:** 3
**Significance:** 2
**Originality:** 2
**Overall Recommendation:** 4
**Confidence:** 3

**Summary:**

This paper proposes Adaptive Sequential Monte Carlo (ASMC) for test-time scaling of LLM reasoning, replacing serial Metropolis-Hastings chains with parallel particle populations targeting power-shaped trajectory distributions. The key systems contribution is cache-coherent resampling, which implements SMC ancestry updates as gather operations over KV cache tensors, avoiding expensive prefix recomputation. On MATH500 with Qwen2.5-Math-7B, ASMC achieves 80.6% accuracy with p95 latency of 73.7s at 128× budget, compared to 73.8% / 1317.5s for sequential MCMC. The paper also introduces adaptive compute allocation (early exit for easy problems, population escalation for hard ones) and provides particle collapse diagnostics showing that low ESS_min/N strongly predicts failures.

**Compliance With Llm Reviewing Policy:**

Affirmed.

**Final Justification:**

I maintain my score since the rebuttal resolves my concerns.

**Key Questions For Authors:**

**Question No.1** What is the accuracy of best-of-n with majority voting at n=8 and n=16 on MATH500? If majority voting eliminates the n=4 saturation, the core Pareto frontier claim would need substantial revision.

**Question No.2** Can you provide multi-seed (≥3) standard deviations and paired statistical tests for the key comparisons in Table 1, particularly ASMC vs best-of-n at 128× budget? The current 1.4% gap is smaller than the expected standard error on 500 binary outcomes at 80% accuracy.

**Question No.3** Is cache-coherent resampling compatible with grouped-query attention (GQA) and paged attention backends such as vLLM? If the gather operation assumes contiguous KV layout with standard multi-head attention, the deployment applicability claimed in the paper would be significantly narrower than stated.

**Limitations:**

This paper does not have a separate limitation section.

The paper briefly discusses limitations within Section 8 (Discussion), acknowledging that conclusions are based on a fixed model and runtime setting, and that relative throughput of prefill and decode can vary with hardware, kernels, and serving concurrency. However, it does not adequately acknowledge the narrow evaluation scope (single model, single main benchmark) as a core limitation, nor does it discuss compatibility constraints with production inference frameworks using paged attention or GQA architectures.

**Strengths And Weaknesses:**

**Strengths**

**Strength No.1:** Cache-coherent resampling is a well-motivated systems primitive with clean microbenchmark evidence showing 53 to 77× latency reduction over prefix replay, and it is strictly necessary for feasibility at high particle counts where rebuild causes OOM. This contribution has reuse value beyond the specific ASMC method.

**Strength No.2:** The deployment-oriented evaluation framework that jointly reports wall-clock p95 latency and compute proxies correctly addresses the known miscalibration between prefill and decode throughput. This dual-reporting practice sets a useful methodological precedent for the test-time scaling literature.

**Strength No.3:** The particle collapse analysis in Section 6.3 provides actionable diagnostics: ESS_min/N strongly predicts failure (p < 1e-15) while the resampling scheme choice has negligible effect, directly guiding practitioners on where to focus optimization effort.

**Weaknesses**

**Weakness No.1: Single model, single primary benchmark.** All main results use only Qwen2.5-Math-7B on MATH500, which cannot support the paper's general claims about test-time scaling. Without validation across different model families, scales, and reasoning domains, the contribution's scope remains unverified.

**Weakness No.2: The accuracy advantage over best-of-n is marginal and statistically unvalidated.** The 1.4% gap (80.6% vs 79.2%) at 128× budget falls within the standard error for 500 problems, and Table 1 reports no confidence intervals or multi-seed variance. Furthermore, best-of-n uses only length-normalized log-probability selection and reportedly saturates at n=4 without explanation, raising concerns that a stronger selection mechanism (majority voting or reward model reranking) would close the gap entirely.

**Weakness No.3: Missing comparison with parallel-friendly test-time scaling alternatives.** The paper does not compare against MCTS-based reasoning, beam search variants with process reward models, or other SMC formulations for LLM inference. Without these baselines, it is unclear whether ASMC's Pareto improvement stems from the specific method design or simply from parallelization over the serial MCMC baseline.

---

> ### Author Rebuttal · Authors · 2026-03-31
>
> We thank Reviewer for the detailed assessment and for highlighting the reuse value of cache-coherent resampling, the latency-aware evaluation, and the collapse diagnostics.
>
> **Q1. Best-of-n with majority voting.**
>
> We ran majority voting on all **500 MATH500 problems** under the same sampling setup as the paper. Using **16 independent samples** per problem and subsampling to smaller n, we obtain the following accuracies: **n = 4: 75.8%**, **n = 8: 79.2%**, and **n = 16: 80.0%**. We will add these results in the revision.
>
> These results do not change the main systems conclusion. Even with stronger aggregation, independent-sampling baselines must wait for all samples to finish, so tail latency is still governed by the slowest trajectory. In our **n = 16 majority-vote** run, **p95 = 916s**, whereas **ASMC-adaptive** reaches **80.6%** at **p95 = 73.7s** on MATH500. We therefore do not claim a large accuracy win over all Best-of-n variants. The intended claim is that ASMC reaches comparable high-budget accuracy while avoiding the extreme tail-latency growth of independent-sampling baselines, and especially sequential MCMC.
>
> **Q2. Statistical support for ASMC vs. Best-of-n.**
>
> We agree that the **1.4 percentage point** gap at **128x** on MATH500 should not be overstated. Appendix Table A1 already reports **95% confidence intervals**: **ASMC-adaptive = 80.6% [77.0, 84.0]** and **Best-of-n = 79.2% [75.6, 82.6]**, so the intervals overlap. We therefore do **not** claim a statistically significant accuracy improvement over Best-of-n from this result alone. We will add multi-seed variance and paired testing in the revision as available. Our intended claim is a **Pareto / latency claim**, not a definitive accuracy-superiority claim over Best-of-n. The decisive comparison is against sequential MCMC, where ASMC improves both accuracy and p95 substantially.
>
> **Q3. GQA and paged-attention backends.**
>
> We have not benchmarked cache-coherent resampling on GQA or paged-attention backends, so we do not want to overclaim. Structurally, the operation acts along the **particle / batch dimension**, not the head dimension, so it should remain compatible with GQA at the algorithmic level. For paged-attention backends, the analogous operation would more likely be **page-table remapping** or **ancestor-indexed metadata updates**, rather than contiguous tensor copying. This remains a systems hypothesis rather than a validated result, and we will state this limitation explicitly in the revision.
>
> **C1. Single model / single primary benchmark.**
>
> We agree that this is an important limitation and will state it more explicitly. To broaden scope, we have now also run a **compute-matched GSM8K evaluation** on **1319 problems** under the same budget-selection protocol. At **8xC0**, ASMC reaches **90.8%** with **p95 = 13.5s**, compared to MCMC at **84.9%** with **p95 = 151.9s**, and Best-of-2 at **51.8%** with **p95 = 24.5s**. At **32xC0**, ASMC reaches **90.5%** with **p95 = 16.5s**, compared to MCMC at **85.5%** with **p95 = 413.8s**, and Best-of-8 at **57.1%** with **p95 = 25.4s**. We will include these results in the revision. Together with HumanEval, this broadens the evaluation to **three benchmarks** spanning math, arithmetic word problems, and code generation. We still agree that the single-model scope remains a limitation.
>
> **Weakness 2 and 3. Framing relative to Best-of-n and other parallel-friendly inference methods.**
>
> We agree that the paper should be framed more carefully relative to Best-of-n, and more broadly relative to other parallel-friendly inference methods. Best-of-n remains a strong baseline, especially at low budgets, and the paper should not be read as claiming a statistically secure quality win over it. We will revise the framing accordingly and broaden the related-work discussion to clarify that our empirical scope is a strict **base-model-only / no-auxiliary-model** setting.
>
> We thank the reviewer again. We believe these clarifications strengthen the paper’s scope and positioning without changing its main takeaway: **cache-coherent resampling is a reusable systems primitive**, and **ASMC makes high-budget particle inference practical by avoiding the extreme tail-latency pathology of sequential chain-based methods**.

---

> > ### Author Rebuttal · Reviewer_9ckj · 2026-04-03
> >
> > Dear authors
> >
> > Thank you for your reply.
> >
> > You can do some revision in your Figure 1 and Figure 2, which would be helpful to all readers. Good work.
> >
> > Your rebuttal makes sense, I keep my original score.
> >
> > Best Regards
> >
> > Reviewer 9ckj

---

### Official Review · Reviewer_Ci4W · 2026-03-11

**Soundness:** 3
**Presentation:** 2
**Significance:** 3
**Originality:** 3
**Overall Recommendation:** 4
**Confidence:** 3

**Summary:**

The authors propose an adaptive, KV cache-aware variant of Sequential Monte Carlo for
inference-time reasoning in Transformer-based models. Motivated by KV cache reuse patterns in Transformer inference, the authors propose a straightforward “cache-coherent state reordering” procedure which
effectively reduces p95 latency on reasoning benchmarks. Common failure modes are examined, which motivate diagnostic methods to mitigate their impact.

**Compliance With Llm Reviewing Policy:**

Affirmed.

**Key Questions For Authors:**

Did ASMC require more GPUs compared with baseline best-of-n methods?

Do the authors foresee inter-GPU communication becoming a bottleneck during cache-coherent state reordering
for large-scale, multi-device models?

Why do the compute metrics in Table A.1. not appear to scale with the compute budget?

**Limitations:**

Yes

**Strengths And Weaknesses:**

Strengths:

1. The experimental results demonstrate that ASMC is highly effective in reducing tail latencies compared to existing Monte Carlo methods. The latency improvements over naive, full-replay MCMC is
particularly notable. Additionally, the heuristic methods for detecting failure modes are well-supported
by experiments (shown in Fig. 4b).

2. Aside from the introduction (see the next section), the paper is well-written and relatively easy to
follow.

3. The authors suitably motivate each of their contributions: the KV cache reuse, the adaptive aspects
of ASMC, the early exit and failure detection, etc. Each subroutine has sufficient explanation, and
it is clear that the authors have produced technically meaningful contributions for Transformer-based
Monte Carlo methods

Weaknesses:

1. The introduction is fairly technical and not particularly accessible to a broad audience. For instance,
it is difficult to extract the setting of the paper (Transformer-based models with reasoning) from the
initial paragraph. The authors immediately jump into discussing existing methods without firmly
establishing the problem they address. Similarly, key metrics used throughout the paper (p95 and
ESS) are not explicitly defined. On a more nit-picky note, there’s a typo in the third paragraph, “.
runs in a sequential decode loop and does not exploit batching” seems to be a remnant of earlier edits.

2. The authors do not seem to directly address the increased compute requirements required by ASMC.
The potential increase in compute overhead is discussed in their Impact Statement, however there
do not seem to be any quantitative comparisons which account for the increased “FLOP”-budget.
Looking at the N counts in Tables 1 and A.1., ASMC seems to require up to a 16x increase in model
instances over simpler methods such as “best-of-n”. However, the compute metrics “Cstep” and “Cint”
do not seem to scale commensurately. It is therefore unclear whether the proposed metrics capture
the increased computational demand, particularly if the authors had sufficient GPUs available to run
the model instances in parallel, which would not affect the key latency metrics. It would therefore be
helpful if the authors could clarify this point.

3. It would be useful if the authors could discuss the empirical/theoretical benefits/tradeoffs of ASMC
within the broader context of inference-time reasoning methods, not just sequential chain MCMC.
In both the introduction and conclusion, the authors focus on the gains over traditional MCMC, which are impressive. However, it may be worth mentioning the comparison (e.g., accuracy/latency
improvements) to best-of-n as well, since the authors mention that as a common paradigm.

Overall, the proposed ASMC method represents a substantial improvement over existing SMC/MCMC algorithms for reasoning in Transformer models. The empirical claims made are supported by extensive experiments. The reviewer therefore thinks that this work merits publication in ICML. That being said, the reviewer would like the authors
to make the introduction more broadly accessible and provide clarification on the hardware requirements of
ASMC compared with baseline methods before providing a full recommendation for publication.

---

> ### Author Rebuttal · Authors · 2026-03-31
>
> We thank Reviewer for the supportive assessment. We are especially encouraged by the reviewer’s view that the method represents a substantial improvement over existing Monte Carlo methods for Transformer-based reasoning. We understand the main remaining concerns to be accessibility of the presentation and clarity about compute and hardware requirements, and we will address both directly in the revision.
>
> **C1. Introduction accessibility, definitions, and typo.**
>
> We agree that the introduction can be made more accessible. In the revision, we will begin with the practical setting, namely improving Transformer-based LLM reasoning at inference time without retraining, before discussing trajectory-level Monte Carlo methods. We will also define **p95** and **ESS** at first use, rather than later in the paper, and we will fix the typo in the third paragraph of the introduction.
>
> **C2 / Q1 / Q3. Compute requirements, hardware setup, and Table A1.**
>
> ASMC did not require additional GPUs or separate model replicas relative to the baselines. All experiments were run in the same **A100 SXM 80GB** environment under fixed hardware and runtime settings. In ASMC, particles are represented as batched states sharing one set of model weights, similar in spirit to how Best-of-n batches multiple samples in one forward pass. The added burden is per-particle KV state plus reweighting and resampling logic, not additional copies of the model.
>
> The reason Table A1 does not scale linearly with the budget cap is that the table reports the **realized mean compute of the selected configuration**, not the cap itself. Our protocol selects the best configuration under **E[C_int] <= 1.02 x C_int_star**, but the selected method does not have to consume the full cap in expectation. This is especially true for adaptive ASMC, where easy problems may exit early and only harder problems escalate. Therefore, the realized mean C_int can grow sublinearly with the cap.
>
> More broadly, **C_int** is an algorithmic workload proxy, not a direct measure of physical GPU count or hardware parallelism. This is exactly why we use measured **wall-clock p95 latency** as the primary deployment-facing metric and use **C_int** only for compute-matched selection. We will make this distinction much more explicit in the revision.
>
> **C3. Broader context, including Best-of-n.**
>
> We agree that the paper should position ASMC not only relative to sequential MCMC, but also relative to Best-of-n, since Best-of-n is an important practical baseline. On MATH500, this comparison is nuanced. At lower budgets, Best-of-n can be a strong practical choice. At higher budgets, ASMC becomes more attractive because Best-of-n largely saturates while sequential MCMC becomes impractical due to extreme tail latency. We will revise the introduction and discussion to make this positioning more explicit.
>
> To strengthen this broader context, we have now run an additional **compute-matched GSM8K evaluation** on **1319 problems** under the same selection protocol. At **8xC0**, ASMC reaches **90.8%** with **p95 = 13.5s**, compared to MCMC at **84.9%** with **p95 = 151.9s**. At **32xC0**, ASMC reaches **90.5%** with **p95 = 16.5s**, compared to MCMC at **85.5%** with **p95 = 413.8s**. We will add these results to the revision. Together with HumanEval, this broadens the evaluation beyond a single primary benchmark.
>
> We do not intend the weak GSM8K Best-of-n result to imply that all Best-of-n variants are weak on GSM8K. In our paper, Best-of-n is intentionally restricted to the same **base-model-only** interface used throughout, namely independent sampling followed by **length-normalized model log-probability reranking**, with no verifier, PRM, or task-specific reward. Under this unified interface, independent sample-and-select correlates poorly with GSM8K exact-match correctness, whereas trajectory-level methods benefit more from reallocating compute during generation.
>
> **Q2. Inter-GPU communication for large multi-device models.**
>
> We have not benchmarked ASMC in a multi-device setting, so we do not want to overclaim. Algorithmically, cache-coherent resampling is an ancestor-indexed gather over the particle dimension for KV caches and other particle-bound tensors. For tensor-parallel models, the same logic should apply shard-locally, but the realized efficiency at larger scales will depend on cache layout, kernel support, and communication overhead. We therefore view inter-GPU communication as an important open systems question and will state this limitation more explicitly in the revision.
>
> We thank the reviewer again for the encouraging assessment. We believe the clarified introduction, the explicit hardware and compute discussion, and the added compute-matched GSM8K results address the main issues raised and will improve the paper substantially.

---

> > ### Author Rebuttal · Reviewer_Ci4W · 2026-04-03
> >
> > Thank you for the detailed and thoughtful response. These clarifications, especially on computing and positioning, are helpful. I appreciate the planned revisions, and my overall assessment and recommendation remain unchanged.

---

> > > ### Author Response · Authors · 2026-04-03
> > >
> > > Thank you for the thoughtful follow-up. We are glad the clarifications were helpful, and we will incorporate them more clearly in the revision.
> > >
> > > If there are any specific remaining questions, we would be happy to clarify them further. Thank you again for your careful reading.

---

### Official Review · Reviewer_CAEt · 2026-03-11

**Soundness:** 2
**Presentation:** 2
**Significance:** 2
**Originality:** 2
**Overall Recommendation:** 4
**Confidence:** 3

**Summary:**

The paper studies test-time scaling for LLM reasoning by replacing sequential Metropolis-Hastings style trajectory sampling with a parallel Sequential Monte Carlo approach. The main proposal, ASMC, uses a population of particles to approximate a power-shaped trajectory distribution, and its key systems contribution is cache-coherent resampling, which updates particle ancestry by reordering KV caches instead of rebuilding prefixes.

**Compliance With Llm Reviewing Policy:**

Affirmed.

**Final Justification:**

Thanks to the authors for the rebuttal discussion.

The added GSM8K results and the ESS threshold ablation partially address our concerns about empirical scope and sensitivity. Also appreciate the more honest framing that the main contribution is systems efficiency rather than a large accuracy gain over best-of-n.

However, we want to push back on C2. The justification for excluding TSMC (Zhao et al., 2024), DisCiPL (Grand et al., 2024), and Rollout Roulette (Puri et al., 2024) was that the paper restricts to "base-model-only, no-extra-training methods." Methods like Rollout Roulette are training-free inference procedures. Having these added baselines would further strengthen the final submission.

I will raise my score accordingly.

**Key Questions For Authors:**

1. How sensitive are the adaptive ASMC results to the ESS threshold, confidence threshold, annealing schedule, block size, and Nfast/Nhard choices? The current method looks fairly heuristic.
2. Since particle collapse is identified as the main failure mode, what concrete interventions did you try beyond comparing residual vs multinomial resampling? Maybe something like diversity-promoting proposals, or adaptive resample criteria?
3. On HumanEval, ASMC improves pass@1 only at high budgets and with much higher latency than best-of-n. Does this suggest the method is mainly suitable for long-horizon math-style reasoning rather than general reasoning or code tasks?

**Limitations:**

yes

**Strengths And Weaknesses:**

Strengths:

1. The paper correctly diagnoses that chain-based MCMC is fundamentally at odds with GPU parallelism, and that KV cache management is a real engineering barrier for particle-based methods in Transformers
2. The paper is commendably transparent about the limitations of C_int as a proxy, and pairs it with wall-clock latency. The dual-proxy framework (C_int + C_step) and the forward-pass hook instrumentation are well-designed.
3. The empirical finding that resampling frequency strongly predicts failure while resampling scheme choice has limited effect is a useful diagnostic insight for the community.

Weaknesses:

1. The main evidence is essentially one base model and one primary task family. ASMC helps only at high budgets and with much worse latency than best-of-n. More empirical support is needed to strengthen the claims.
2. The comparisons are mainly against greedy, best-of-n, and one sequential MCMC baseline. For a paper making strong claims about efficient test-time scaling. There are other relevant work like TSMC (Feng et. al), Rollout Roulette (Puri et. al), DisCiPL (Grand et. al), etc.
3.  The paper explicitly says interaction-based compute can be miscalibrated, which is fair, but then still uses that proxy for compute-matched selection. Why?
4.  The early-exit and fast-to-hard escalation rules seem plausible, but they look heuristic. I did not see enough ablation showing how sensitive results are to the thresholds or whether the adaptive gains are robust.
5. On MATH500, the strongest headline at 128x is roughly 80.6 vs 79.2, while the bigger win is latency. That is still useful, but the paper should be more explicit that the main advantage is systems efficiency under large budgets, not a large jump in reasoning quality.

---

> ### Author Rebuttal · Authors · 2026-03-31
>
> We thank Reviewer CAEt for the careful and constructive review. We appreciate both the positive assessment of the systems motivation and the concern that the original submission was empirically too narrow.
>
> **C1. Limited empirical scope; “ASMC only helps at high budgets and with worse latency than Best-of-n.”**
>
> We agree that the paper’s main contribution is systems-efficient test-time scaling for Transformer reasoning, not a claim of uniformly higher raw accuracy than every baseline. We will make this framing explicit in the revision. The strongest controlled result in the current submission is on MATH500: adaptive ASMC reaches 80.6% at  p95 = 73.7s, versus 73.8% at p95 = 1317.5s for sequential MCMC. The gap over Best-of-n is much smaller, and we will state that more clearly.
>
> To address the concern that our evaluation was too narrow empirically, we have now conducted a full compute-matched evaluation on GSM8K under the exact same budget-selection protocol as the main paper. At 8xC0, ASMC (N=8, T_ann=256) reaches 90.8% with  p95 = 13.5s, versus 84.9% and 151.9s or MCMC. At 32xC0, ASMC (N=64, T_ann=1024) reaches 90.5% with p95 = 16.5s, versus 85.5% and 413.8s for MCMC. Under the same base-model-only interface, Best-of-n remains much lower in accuracy on GSM8K. We will include these results in the revision. We believe this directly addresses both concerns: the evidence is no longer limited to a single benchmark, and the gains are not confined to only the highest-budget regime.
>
> We also want to clarify the interpretation of the weak GSM8K Best-of-n result. In our paper, Best-of-n is intentionally restricted to the same base-model-only interface used throughout: independent sampling followed by length-normalized model log-probability reranking, with no verifier, PRM, or task-specific reward. We therefore do not interpret this as evidence that all Best-of-n variants are weak on GSM8K.
> The narrower conclusion is that, under this unified interface, sample-and-select correlates poorly with exact-match correctness.
>
> **C2. Missing baselines such as TSMC, Rollout Roulette, and DisCiPL.**
>
> We agree that these methods are relevant. Our experiments were intentionally restricted to **base-model-only, no-extra-training** methods under a common scoring interface, which is why we focused on greedy, Best-of-n, and sequential MCMC. We will revise the scope statement and related work to make this boundary explicit and avoid implying broader empirical coverage than we currently provide.
>
> **C3. Why use C_int if it can be miscalibrated across methods?**
>
> We agree that C_int can be miscalibrated across methods. That is exactly why our main conclusions are based on **wall-clock p95 latency**, not on C_int itself. C_int is used only for **budget-capped configuration selection**, after which methods are compared on realized latency. We will make this two-step protocol clearer in the revision.
>
> **C4 / Q1. Sensitivity to ESS threshold, confidence threshold, annealing schedule, block size, and N_fast / N_hard.**
>
> We agree that the adaptive policy is heuristic. Our claim is not that these thresholds are optimal, but that the main systems conclusion does not depend on a narrow choice. We ran a full 500-problem sweep over the ESS threshold tau on MATH500 at adaptive **N=64** and **T_ann=1024**. Accuracy changes only slightly across **tau in {0.3, 0.5, 0.7}**: **79.4 / 79.2 / 79.0**. We will add this ablation and present the other adaptive knobs more clearly as heuristic efficiency layers.
>
> **Q2. What interventions were tried beyond residual vs. multinomial resampling?**
>
> Beyond residual vs. multinomial resampling, the current method already includes a defensive mixture proposal and ESS-triggered resampling to reduce unnecessary degeneracy. We agree, however, that these are not yet stronger anti-collapse interventions in the sense of rejuvenation-style moves. We have not yet implemented stronger interventions such as rejuvenation moves in the current submission; instead, our current evidence is diagnostic: low ESSmin/N strongly predicts failure, while switching standard resampling schemes has only a small effect. This is why we identify stronger rejuvenation and diversity-preserving moves as the most important next step.
>
> **Q3. Does HumanEval suggest the method is mainly suitable for long-horizon math-style reasoning?**
>
> We agree that the current evidence most strongly supports **verifiable reasoning tasks**. MATH500 remains the clearest demonstration, HumanEval shows a weaker latency-quality tradeoff, and the new GSM8K results suggest the method is not specific to a single math benchmark. We will state this scope boundary more explicitly in the revision.
>
> We hope that the added GSM8K results, the clearer framing relative to Best-of-n, the explicit scope clarification on baselines, and the added sensitivity evidence address the main reasons behind the current recommendation, and we would be grateful for reconsideration.

---

> > ### Author Rebuttal · Reviewer_CAEt · 2026-04-02
> >
> > Thanks to the authors for the rebuttal discussion.
> >
> > The added GSM8K results and the ESS threshold ablation partially address our concerns about empirical scope and sensitivity. Also appreciate the more honest framing that the main contribution is systems efficiency rather than a large accuracy gain over best-of-n.
> >
> > However, I want to push back on C2. The justification for excluding TSMC (Zhao et al., 2024), DisCiPL (Grand et al., 2024), and Rollout Roulette (Puri et al., 2024) was that the paper restricts to "base-model-only, no-extra-training methods." Methods like Rollout Roulette are training-free inference procedures. Having these added baselines would further strengthen the final submission.
> >
> > I will raise my score accordingly.

---

> > > ### Author Response · Authors · 2026-04-02
> > >
> > > Thank you for the follow-up, and we appreciate your score update.
> > >
> > > We agree that our earlier wording in C2 was too coarse. In particular, Rollout Roulette is a training-free inference baseline, but it is not in the same base-model-only log-probability setting as our paper, since its main experiments use an auxiliary reward model (Qwen2.5-Math-PRM-7B). And TSMC-style methods use learned twist/value-style components, and DisCiPL uses a planner/follower inference-program formulation.
> > >
> > > By contrast, our comparisons are restricted to methods that use only the base model’s log-probabilities at inference time. We will revise the paper to make this taxonomy more precise and to position these baselines more accurately in related work. Thank you again for helping us sharpen this distinction.

---

### Official Review · Reviewer_2xYA · 2026-03-12

**Soundness:** 3
**Presentation:** 2
**Significance:** 4
**Originality:** 3
**Overall Recommendation:** 5
**Confidence:** 3

**Summary:**

This paper addresses the efficiency bottlenecks of "test-time scaling" for Large Language Model (LLM) reasoning. Current methods that improve reasoning performance by sampling from power-shaped trajectory distributions rely heavily on Markov Chain Monte Carlo (MCMC) algorithms, such as Metropolis-Hastings. These approaches are inherently sequential, failing to exploit GPU parallelism and resulting in extreme tail latency (p95) at high compute budgets.

To mitigate this, the authors propose **Adaptive Sequential Monte Carlo (ASMC)**. ASMC replaces serial Markov chains with a parallel population of particles, effectively transforming a temporal search into a spatial search compatible with batched execution. A key technical contribution is the introduction of **"Cache-Coherent Resampling,"** which reorders per-layer KV caches and particle-associated tensors via gather operations. This system-level optimization avoids the $O(NL^2)$ cost of prefix recomputation during resampling events. Additionally, the framework incorporates adaptive compute allocation, dynamically scaling the particle population ($N$) or early-exiting based on problem difficulty and confidence metrics.

Evaluations on MATH500 and HumanEval demonstrate that ASMC achieves a superior Pareto trade-off between accuracy and p95 latency compared to sequential MCMC and Best-of-n baselines, significantly improving the practicality of high-budget test-time scaling.

**Compliance With Llm Reviewing Policy:**

Affirmed.

**Key Questions For Authors:**

1. While the authors include HumanEval in the appendix to show generalization, both MATH and Code are tasks with "hard" verifiers. In the limitations, the authors note the need for "stronger rejuvenation moves" to improve diversity. Can the authors clarify if ASMC’s current reliance on high-confidence "top-mass" early exits would lead to premature collapse in semantic tasks (e.g., summarization or creative reasoning) where multiple valid paths exist but likelihoods are flatter?

2. The authors correctly identify hardware-specific throughput as a limitation, yet they do not discuss the **Memory Bandwidth Bottleneck** of the gather operation itself for large models. As $D_{model}$ scales (e.g., from 7B to 70B+), the ratio of data movement for KV-cache reordering vs. computation changes. Does the author have microbenchmark data or a theoretical upper bound to show at what model size the "Cache-Coherent Reordering" stops being more efficient than a simple prefix replay?

3. Acknowledging that adaptive policies are future work does not mitigate the current lack of a convergence guarantee. Given the discrete nature of LLM token space, how can we be sure that the importance weights $w_t^{(i)}$ don't introduce biased samples under the proposed annealing schedule $\{\alpha_t\}$, especially when $ESS < \tau N$ triggers frequent resampling?

4. Could the authors standardize the color palette (or unification of component styles) across figures to ensure a more unified style throughout the paper? Moreover, the font size of some legends in the figures (especially Figure 2.) are too small to recognize. The modification of their figures will help improve the evaluation of Presentation part.

**Limitations:**

Yes

**Strengths And Weaknesses:**

Soundness: The submission is technically sound, supported by extensive experiments on MATH500 and HumanEval. The use of attention interaction $C_{int}$ paired with p95 wall-clock latency provides a robust, multi-dimensional evaluation of efficiency. The proposed cache-coherent reordering is methodologically appropriate for Transformer architectures, successfully avoiding quadratic recomputation costs. However, the annealing schedule and defensive mixture proposal, while empirically effective, lack a rigorous theoretical proof regarding their optimal convergence to the target power distribution. Additionally, while the $ESS$ threshold is a standard SMC diagnostic, its specific sensitivity to the discrete, high-dimensional nature of token sequences could be more deeply explored. The authors are honest about the failure mode of particle collapse, though the mitigation through "rejuvenation moves" is left for future work.

Presentation: The paper is clearly written with a narrative that logically flows from identifying the serial bottleneck of MCMC to the parallel solution of ASMC. Figure 1 and Algorithm 1 provide a high-quality visualization of the system, making the complex ancestry updates accessible to readers. The authors properly position their work by distinguishing SMC from traditional chain-based MCMC. On the downside, the mathematical derivation of the conditional power distribution in Section 2.1 is somewhat dense, and the distinction between "sum of exponents" and "exponent of sums" could be more intuitively phrased for non-specialists. While Appendix Table A1 is exhaustive, some key hyperparameter sensitivities, like the impact of block size $B$, are buried in the back matter. Improving the labeling of the feasibility heatmaps in Figure 3 could also enhance immediate scannability.

Significance: This work addresses the highly relevant problem of test-time scaling, offering a path to reach high-performance reasoning without expensive retraining. By reducing the p95 latency on MATH500 from 1318s to 73.7s at a 128x budget, the significance for real-world LLM deployment is substantial. The introduction of a hardware-friendly resampling primitive could influence future practitioners to adopt population-based inference over simple Best-of-n. However, the significance is somewhat domain-specific, as the current evaluation centers heavily on math and coding, leaving its broader utility for generalized chat or low-entropy tasks unproven. Furthermore, the performance gains are measured on a 7B model; it remains an open question whether the data-movement overhead of KV cache reordering scales favorably as model size increases significantly.

Originality: The paper demonstrates originality through the creative co-design of a probabilistic inference method and Transformer-specific system optimizations. While SMC and power-shaped distributions are existing concepts, the application to LLM reasoning—specifically solving the ancestry-management bottleneck via GPU gather operations—is a novel and insightful contribution. The adaptive population strategy ($N_{fast}$ to $N_{hard}$) further adds to the originality by introducing a dynamic compute-allocation layer absent in prior MCMC-based scaling works. Conversely, the "cache-coherent" technique, while effective, essentially applies standard tensor reindexing to a new domain, and the underlying resampling schemes are borrowed from classical literature. Despite this, the work successfully removes the restrictive assumption that resampling requires prefix replay, representing a clear advancement.

---

> ### Author Rebuttal · Authors · 2026-03-31
>
> We thank Reviewer for the thoughtful and constructive review, and for highlighting both the practical significance of reducing tail latency and the need to better clarify the scope and guarantees of the method.
>
> **Q1. Premature collapse on semantic tasks with flatter likelihoods.**
>
> We agree this is an important scope question. Our current evidence is strongest for tasks with verifiable outcomes and relatively concentrated reward structure (MATH500 in the main text, HumanEval in the appendix), rather than open-ended semantic generation. In our framework, the top-mass early-exit rule is an optional heuristic layered on top of the core ASMC/SMC procedure, not part of the inference method itself. If early exit is disabled, particles simply continue to the full budget under the same ASMC weighting and aggregation procedure.
>
> More broadly, we agree that when many valid trajectories have similar likelihoods, power-shaped targets may provide smaller gains, and diversity-preserving moves become more important. We will clarify this scope boundary in the revision and state more explicitly that the current method is best matched to tasks with verifiable outcomes, while stronger rejuvenation/diversity mechanisms are likely needed for flatter semantic tasks.
>
> **Q2. Memory-bandwidth bottleneck of gather at larger model scales.**
>
> The key systems comparison is gather-based cache reordering versus prefix replay. In our current 7B setting, cache-coherent reordering yields a large latency advantage over replay and remains feasible in boundary regimes where replay becomes timeout- or memory-limited. However, we do not yet have 70B+ or multi-device microbenchmarks, and we do not want to overclaim validated scaling beyond the current setup.
>
> We will revise the paper to make this limitation explicit. At a high level, the method only requires ancestor-indexed reordering of particle-bound state, but the realized efficiency at larger scales will depend on cache layout, GQA/paged-attention implementation, kernel support, and inter-device communication. Establishing the crossover point where reordering ceases to dominate replay is an open systems question that requires dedicated empirical study.
>
> **Q3. Convergence guarantees under the annealing schedule and adaptive policy.**
>
> We agree that the current paper does not provide a formal convergence theorem for the full adaptive procedure. Standard SMC theory gives consistency results for a fixed sequence of intermediate targets under standard conditions, and our use of defensive mixture proposals and standard resampling follows that classical setting. However, the implemented adaptive layers in this paper, especially early exit, hardness-based escalation, and the chosen annealing policy, are heuristic efficiency mechanisms and are not covered by a formal theorem in the current submission.
>
> We will make this distinction much clearer in the revision:
>
> 1. the classical SMC core follows standard principles for fixed intermediate targets;
> 2. the adaptive layers are empirically motivated and currently justified by experimental behavior rather than theory.
>
> **ESS sensitivity in discrete token spaces.**
> We agree that the behavior of ESS in high-dimensional discrete sequence spaces deserves more analysis. Empirically, we ran a full 500-problem sweep over the ESS threshold on MATH500 at adaptive $N=64$ and $T_{\mathrm{ann}}=1024$, using $\tau \in {0.3, 0.5, 0.7}$. The resulting accuracies vary only slightly (79.4 / 79.2 / 79.0), suggesting that performance is not overly sensitive to this threshold in our current setting. We will add this ablation to the revision and frame a deeper theoretical treatment of ESS in autoregressive discrete spaces as an open direction.
>
> **Presentation improvements.**
> Thank you for these suggestions. We will revise the presentation accordingly by:
>
> 1. adding a more intuitive explanation of “sum of exponents” vs. “exponent of sums”;
> 2. moving key hyperparameter sensitivities, including block size, into the main text;
> 3. improving the labeling and immediate readability of Figure 3 heatmaps;
> 4. standardizing the color palette and increasing legend/font sizes across figures; and
> 5. defining p95 and ESS earlier in the introduction.
>
> We thank the reviewer again for the detailed feedback. We believe these clarifications will improve both the scope framing and the presentation of the paper.

---

> > ### Author Rebuttal · Reviewer_2xYA · 2026-04-02
> >
> > I thank the authors for their detailed response. The authors addressed concerns regarding particle collapse in semantic tasks, memory bottlenecks in large models, and convergence proofs, promising to include ESS sensitivity tests and visualization improvements in the revision.
> >
> > While the rebuttal clarified implementation details, I maintain my score for the following reasons:
> >
> > 1. Theoretical vs. Empirical Boundaries: The authors acknowledge that adaptive layers (e.g., Early Exit) are primarily heuristic and lack formal convergence proofs in discrete token spaces, limiting the framework's theoretical depth.
> > 2. Scalability Uncertainty: Micro-benchmark data for the performance crossover point between KV-cache reordering and prefix replay on 70B+ models is still missing. Given memory bandwidth constraints in large models, the systemic advantage remains to be verified.
> > 3. Task Generalization: ASMC is currently validated on tasks with "hard verifiers" (MATH/Code). Its effectiveness in broader natural language generation (e.g., creative writing) remains an open question.
> >
> > I appreciate the proposed revisions, particularly the intuitive explanations and improved figure legibility. Given that the primary contributions are validated on 7B-scale models for verifiable tasks, I maintain my original score.

---

### Decision · Program_Chairs · 2026-04-30

**Decision:**

Accept (regular)

**Comment:**

This paper introduces ASMC, a cache-coherent SMC method for test-time scaling in LLMs, whose central contribution — a GPU-friendly resampling primitive that avoids prefix replay — achieves significant latency reductions. The work is well-motivated, clearly presented, and the dual evaluation framework sets a useful methodological precedent for the test-time scaling literature. Reviewers noted that the empirical scope is limited to a single model and benchmark, the accuracy gains over best-of-n are marginal, and comparisons against parallel-friendly baselines are missing. The compute overhead relative to simpler methods also warrants clearer quantification. Despite these limitations, the reviewers agree that the core contribution is valuable.